    

# PD-1/LAG-3 co-signaling profiling uncovers CBL ubiquitin ligases as key immunotherapy targets

Luisa Chocarro [ID][1,7 ✉], Ester Blanco[1,2,7], Leticia Fernandez-Rubio[1], Maider Garnica [ID][1], Miren Zuazo[1], Maria Jesus Garcia[1], Ana Bocanegra [ID][1], Miriam Echaide[1], Colette Johnston[3], Carolyn J Edwards[3], James Legg [ID][3], Andrew J Pierce [ID][3], Hugo Arasanz [ID][4,5], Gonzalo Fernandez-Hinojal[4], Ruth Vera [ID][4], Karina Ausin [ID][6], Enrique Santamaria[6], Joaquin Fernandez-Irigoyen [ID][6], Grazyna Kochan [ID][1] & David Escors [ID][1 ✉]

## Abstract

Many cancer patients do not benefit from PD-L1/PD-1 blockade immunotherapies. PD-1 and LAG-3 co-upregulation in T-cells is one of the major mechanisms of resistance by establishing a highly dysfunctional state in T-cells. To identify shared features associated to PD-1/LAG-3 dysfunctionality in human cancers and T-cells, multiomic expression profiles were obtained for all TCGA cancers immune infiltrates. A PD-1/LAG-3 dysfunctional signature was found which regulated immune, metabolic, genetic, and epigenetic pathways, but especially a reinforced negative regulation of the TCR signalosome. These results were validated in T-cell lines with constitutively active PD-1, LAG-3 pathways and their combination. A differential analysis of the proteome of PD-1/LAG-3 T-cells showed a specific enrichment in ubiquitin ligases participating in E3 ubiquitination pathways. PD-1/LAG-3 co-blockade inhibited CBL-B expression, while the use of a bispecific drug in clinical development also repressed C-CBL expression, which reverted T-cell dysfunctionality in lung cancer patients resistant to PD-L1/PD-1 blockade. The combination of CBL-B-specific small molecule inhibitors with anti-PD-1/anti-LAG-3 immunotherapies demonstrated notable therapeutic efficacy in models of lung cancer refractory to immunotherapies, overcoming PD-1/LAG-3 mediated resistance.

**Keywords** Cancer Immunotherapy; CBL Ubiquitin Ligases; LAG-3; PD-1; T-cell Dysfunctionality
**Subject Categories** Cancer; Chromatin, Transcription & Genomics; Immunology

## Introduction

PD-L1/PD-1 immune checkpoint (IC) blockade has revolutionised oncology (Topalian et al, 2015; Topalian et al, 2016). This strategy reactivates cancer-specific T-cells leading to tumor regression or long-term disease control. However, intrinsic resistance to PD-L1/PD-1 blockade frequently occurs through several mechanisms. One of the most relevant is the co-expression of LAG-3 with PD-1 in T-cells, which establishes strong T-cell dysfunctionality in patients leading to resistance to conventional immunotherapies, as we and others have shown (Chocarro et al, 2022a; Zuazo et al, 2020; Zuazo et al, 2019). T-cell dysfunctionality was partially overcome ex vivo with PD-1 and LAG-3 co-blockade in T-cells from non-small cell lung cancer (NSCLC) patients, which recovered proliferative and effector activities (Edwards et al, 2022; Zuazo et al, 2019). Hence, PD-1 and LAG-3 cooperatively establish a strong dysfunctional state in T-cells possibly through co-signaling. The mechanisms by which PD-1 interferes with T-cell functions are well-known and reviewed elsewhere (Arasanz et al, 2017). Briefly, PD-1 signaling terminates T-cell receptor (TCR) signal transduction, causes its down-modulation and degradation, and impacts on T-cell metabolism (Chemnitz et al, 2004; Hui et al, 2017; Karwacz et al, 2011; Patsoukis et al, 2015; Patsoukis et al, 2013; Plas et al, 1996; Sheppard et al, 2004). However, the specific inhibitory mechanisms regulated by LAG-3, and the combined effects of PD-1 and LAG-3 co-signaling are largely unknown (Edwards et al, 2022; Zuazo et al, 2019). It has been suggested that PD-1 and LAG-3 establish a complex together with TCR components, that would be required for their combined inhibitory activities (Chen and Flies, 2013; Hannier et al, 1998; Iouzalen et al, 2001; Lichtenegger et al, 2018; Saito et al, 2010).

The clinical significance of PD-1/LAG-3 co-blockade has been recently demonstrated by the approval of Opdualag, a therapy combining anti-PD-1 (nivolumab) and anti-LAG-3 (relatlimab)

[1]OncoImmunology Unit, Navarrabiomed - Fundación Miguel Servet, Universidad Pública de Navarra (UPNA), Hospital Universitario de Navarra (HUN), Instituto de Investigación Sanitaria de Navarra (IdiSNA), 31008 Pamplona, Spain. [2]Division of Gene Therapy and Regulation of Gene Expression, Cima Universidad de Navarra, Instituto de Investigación Sanitaria de Navarra (IdiSNA), 31008 Pamplona, Spain. [3]Crescendo Biologics Ltd., Meditrina Building, Babraham Research Campus, Cambridge CB22 3AT, UK. [4]Medical Oncology Unit, Hospital Universitario de Navarra (HUN), Instituto de Investigación Sanitaria de Navarra (IdiSNA), 31008 Pamplona, Spain. [5]Oncobiona Unit, Navarrabiomed, Instituto de Investigación Sanitaria de Navarra (IdiSNA), 31008 Pamplona, Spain. [6]Proteomics Platform, Proteored-ISCIII, Navarrabiomed - Fundación Miguel Servet, Universidad Pública de Navarra (UPNA), Hospital Universitario de Navarra (HUN), Instituto de Investigación Sanitaria de Navarra (IdiSNA), 31008 Pamplona, Spain. [7]These authors contributed equally as first authors: Luisa Chocarro, Ester Blanco. ✉E-mail: lchocard@navarra.es; descorsm@navarra.es

antibodies. This strategy doubled median progression-free survival of melanoma patients compared to PD-1 blockade alone (Chocarro et al, 2022a; Paik, 2022). However, it is unclear whether all tumor types may benefit from this combination. Although much has been discovered on PD-1 functions and its blockade, the molecular effects of PD-1/LAG-3 co-blockade over PD-1/LAG-3-associated dysfunctional signatures in cancer cells and T-cells are largely unknown. The identification of PD-1/LAG-3 co-signaling gene and protein signatures in T-cells could uncover potential key targets that regulate this strongly inhibitory co-signaling route, which could be pharmaceutically targeted to improve anti-PD-1/anti-LAG-3 combination immunotherapies.

Here we have first systematically analyzed extensive multiomic data to extract PD-1 and LAG-3 omic signatures in all TCGA cancers to compare with PD-1/LAG-3-associated signatures in T-cell lines with constitutive PD-1 and LAG-3 signaling. PD-1/LAG-3 co-signaling profiles uncovered a regulated, highly dysfunctional programme which strongly interfered with the TCR signalosome and was enriched in ubiquitin ligases known to regulate TCR expression and signaling. Co-blockade with anti-PD-1/anti-LAG-3 antibodies in primary human T-cells from NSCLC patients prevented the expression of CBL-B ubiquitin ligases. The use of a PD-1/LAG-3 bispecific inhibitor also further inhibited C-CBL, resulting in the efficacious reversal from T-cell dysfunctionality. The combination of CBL-B-specific small molecule inhibitors with anti-PD-1/anti-LAG-3 antibody immunotherapies resulted in significant therapeutic efficacies in lung cancer models resistant to conventional immunotherapies.

## Results

### PDCD1/LAG3 tumor signatures contain a gene expression profile indicative of highly dysfunctional T cells

Our study aimed to address three major goals. The first one was to extract PD-1/LAG-3 gene co-expression signatures from human cancers using multiomic data associated to T-cell functions. The second goal was to specifically co-activate PD-1 and LAG-3 signaling pathways in T-cells to identify the regulatory interactomes and compare them with the extracted signatures from human omic data. The third goal was to select a protein target associated to the regulation of central T-cell dysfunctional pathways to improve PD-1/LAG-3 immunotherapies by its pharmaceutical inhibition.

Hence, to define gene signatures associated to PDCD1/LAG3 co-expression in human cancers, we performed an extensive analysis of publicly available multiomics cancer data. Public transcriptomic and genomic databases for all TCGA human cancers were analyzed with TIMER 2.0, Gene Expression Profiling Interactive Analysis (GEPIA), cBioportal, TNM Plot and Ingenuity Pathway Analysis (IPA). Overall, this study encompassed more than 12000 patients distributed into 40 cancer types. As a first approach, we tested whether PDCD1 and LAG3 gene expression in tumor-immune infiltrate estimates correlated with immune cell infiltration by lymphoid (Fig. EV1A) and myeloid (Fig. EV1B) lineages. To achieve this, TIMER 2.0 was used to calculate correlation between immune checkpoint expression and the degree of infiltration by

lymphoid and non-lymphoid cell infiltrates using purity-adjusted Spearman's tests (Fig. EV1). PDCD1 and LAG3 positively correlated with infiltrates of T-cells at several differentiation stages, gamma/delta T-cells, activated natural killer cells (NKs), and naïve, memory and class-switched memory B cells. In contrast, PDCD1 and LAG3 signatures negatively correlated with infiltration by naïve non-regulatory CD4 T-cells, resting NK cells, and common lymphoid progenitors. Overall, PDCD1 and LAG3 signatures in human tumors were associated with an inflamed microenvironment signature represented by T-cell and myeloid cell infiltration (Fig. EV1).

Then, PDCD1 and LAG3 gene expressions were correlated with all gene data available in the TIMER 2.0 TCGA data from which we evaluated a selection encompassing 366 genes of T-cell function, 112 interleukin and interferon family genes, and other 254 immune-related genes. A positive correlation with most interleukin and interferon genes was found in tumor infiltrates genes by TIMER 2.0 (Fig. 1A,B; Dataset EV1). As expected, the PDCD1/LAG3 signature correlated with regulators of TCR signaling, CD28 family of costimulation and MHC II antigen presentation molecules, multiple HLA genes, among others. To identify the functional pathways associated to these immune genes, these were reconstructed singly and also in combination as regulatory protein networks and causal relationships with IPA (Kramer et al, 2014) (Figs. 1C and EV2). The resulting interactomes were indicative of T-cell exhaustion pathways, PD-1/PD-L1 blockade immunotherapy and inositol metabolism, all linked to TCR signaling (Fig. EV2A). Causal relationships in these networks were identified to specific signaling and antigen-presentation molecules (Fig. EV2A). A cascade of upstream transcriptional regulators was found associated to the PD-1/LAG-3 signature (Fig. EV2A). The PD-1/LAG-3 co-signature was linked to molecular and cellular functions, and diseases characteristic of immune dysfunctionality and cancer (Fig. EV2B,C). Individual interactome networks for activated PD-1 and LAG-3 pathways or their combination were modelled (Fig. EV2D), and the correlation of the identified molecules with PDCD1/LAG3 expression was validated on human cancers with TIMER2.0 (Fig. EV2E). Some of these were well-known T-cell inhibitors and exhaustion mediators activated by PD-1 signaling such as PTPN22, SATB1 and FOXP3 among others. All the information for the combination was integrated in a PD-1/LAG-3 co-expression signature network which highlighted reinforced T-cell exhaustion pathways leading to downregulation/inhibition of the TCR signalosome (Fig. 1C).

Then, the PDCD1/LAG3 gene expression profile was further correlated with other available tumor markers and oncogenic processes, which included an additional 1038 non-immune-related genes, regulators of cell cycle, cell-cell communication, chromatin organization, DNA repair, DNA replication, programmed cell death and autophagy (Fig. 1D). Most of the studied genes showed strong correlation with cancer-altered gene regulation profiles. Interestingly, an association was found with downstream targets which further negatively regulated the TCR signalosome, including genes involved in immune synapse formation, TCR-associated kinases, phosphatases, cytokine signaling kinases and PI3K/AKT signaling pathway, among others (Dataset EV2).

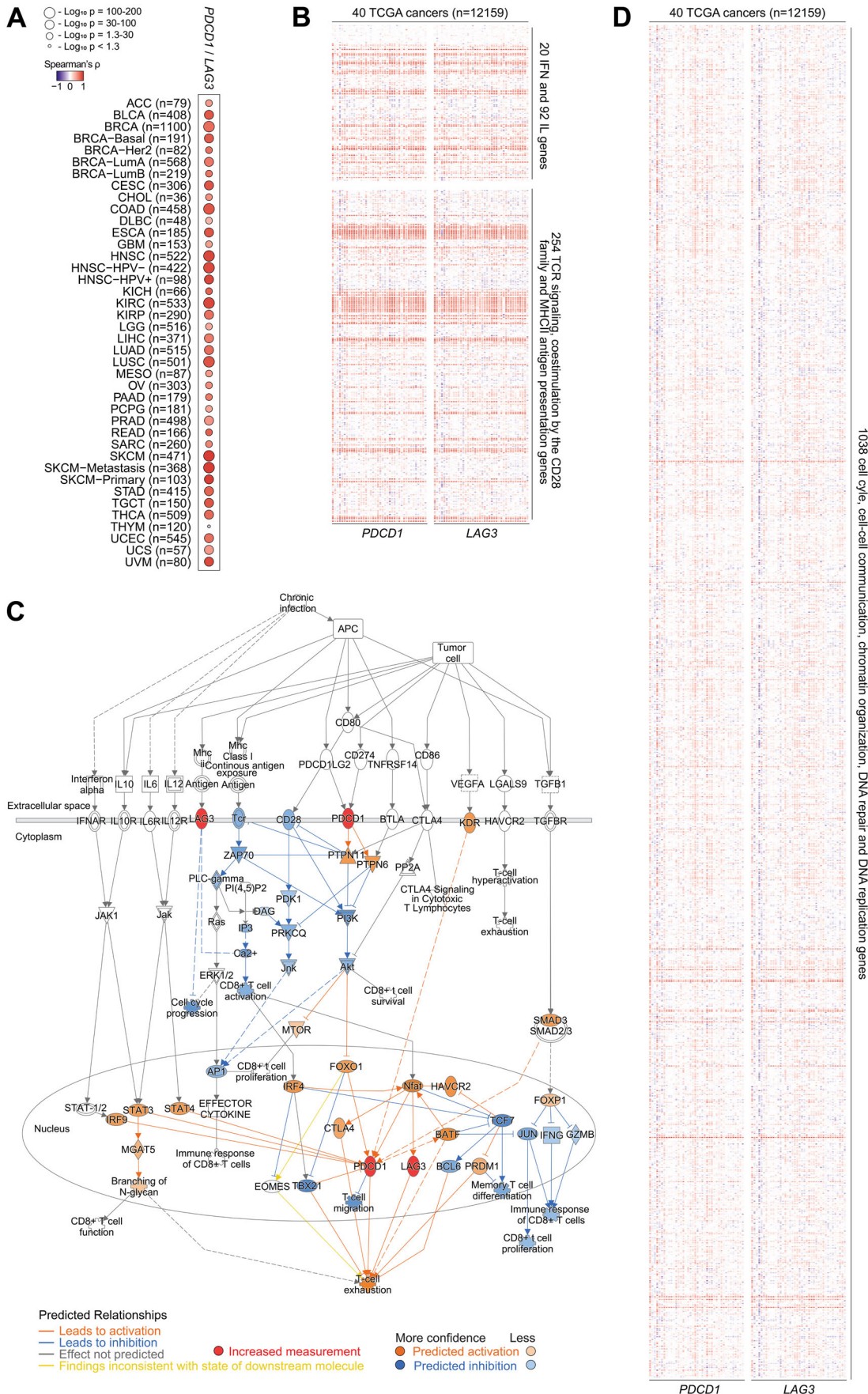

◄ **Figure 1.    Identification of the *PDCD1/LAG3* tumor-infiltrate signature on human cancers.**

(A) Heatmap of partial purity-adjusted Spearman's correlates calculated with TIMER 2.0. between *PDCD1* and *LAG3* co-expression in the tumor-immune infiltration estimation of a total number of 12159 samples distributed on the indicated TCGA cancers. (B) Heatmap of partial purity-adjusted Spearman's correlates calculated with TIMER 2.0. between *PDCD1/LAG3* expression and selected immune genes including IFN, IL, TCR signaling, CD28 co-stimulatory family and MHCII antigen presentation in the tumor-immune infiltration estimation of a total number of 12159 samples distributed on the indicated TCGA cancers. Detailed information and statistical significance for each specific gene associated to the *PDCD1/LAG3* gene signature is shown in Dataset EV1. (C) Predicted network describing the potential molecular interactions for the T-cell exhaustion pathway signaling and TCR downregulation associated to both PD-1 and LAG-3 co-upregulation. QIAGEN IPA algorithms were applied on data from curated publicly available datasets of RNA-seq, small RNA-seq, metabolomics, proteomics, microarrays including miRNA and SNP, and small-scale experiments (accessed on 2024). The specific legends to inter-nodal relationships are described in IPA (Ingenuity Pathway Analysis | QIAGEN Digital Insights). Key nodes are shown, and inter-nodal lines represent potential functional relationships between nodes. In red, upregulated input molecules as indicated (PD-1 and LAG-3). Blue lines, predicted inhibition; orange lines, predicted activation; grey indicates a predicted relationship with a non-predicted effect, and yellow lines, predicted relationship findings inconsistent with the state of the downstream molecule. (D) Heatmap of partial purity-adjusted Spearman's correlates calculated with TIMER 2.0. between *PDCD1/LAG3* expression and a selection of genes regulating cell cycle, gene expression and signaling in the tumor-immune infiltration estimation of a total number of 12,159 samples distributed on the indicated TCGA cancers. Detailed information and statistical significance for each specific gene associated to the *PDCD1/LAG3* gene signature is shown in Dataset EV2. Source data are available online for this figure.

## Constitutive activation of PD-1 and LAG-3 signaling pathways and their combination in T-cell lines

Gene and protein networks indicative of T-cell dysfunctionality could be extracted from *PDCD1/LAG3* omics signatures in human cancers. To compare these signatures with those in T-cells with activated PD-1/LAG-3 signaling pathways, we generated T-cell lines with sustained PD-1/LAG-3 signaling either separately or in combination. To that end, PD-1 and LAG-3 molecules with constitutive signaling activities in T-cells were engineered. PD-1 and LAG-3 extracellular immunoglobulin-like domains were replaced with a CD3-binding single chain antibody (SC3) previously described by us (Zuazo et al, 2019). When expressed in T-cells, the SC3 domain binds to CD3 from neighbouring T-cells. This interaction simultaneously triggers TCR signal transduction and signaling through the intracellular domains of PD-1, LAG-3 or both (PD-1 + LAG-3 cell lines) (Fig. 2A). These fusion genes (termed SC3-PD-1 and SC3-LAG-3) were cloned into lentivectors for T-cell transduction. As a control, the SC3 molecule without PD-1 or LAG-3 intracellular tails was used (Zuazo et al, 2019). To test their expression, GFP and mCherry were fused to their carboxy-termini. Both PD-1 and LAG-3-based constructs showed good expression levels and accumulated at cell-to-cell contacts in human primary cells and Jurkat CD4 T-cells (Fig. 2B,C). However, the viability and growth of primary T-cells when expressing these constructs was highly compromised. The growth of Jurkat T-cells expressing either of the constructs was significantly delayed compared to control cell lines, especially for PD-1 + LAG-3 co-expression, although it was possible to grow them as cell lines (Fig. 2D). The same constructs without GFP or mCherry exerted the same inhibitory effects and were chosen for further studies to eliminate artefacts in omics data caused by the expression of fluorescent proteins.

The effects of PD-1 and LAG-3 signaling singly or in combination (PD-1 + LAG-3) over the T-cell phenotype were studied over the expression of cell markers of T-cell dysfunctionality, proliferation, and DNA damage (Fig. 2E,F). PD-1, LAG-3 and PD-1 + LAG-3 showed distinct phenotypes related to inhibitory functions. A strong inhibition of proliferation was evidenced by reduced Ki67 expression, and key T-cell markers were downregulated (Fig. 2E). ZAP70 expression was evaluated as a marker of TCR signaling capacities, which was strongly reduced by PD-1 or LAG-3 signaling, and nearly completely abrogated by the PD-1 + LAG-3 combination (Fig. 2E). Evidence of accumulated DNA damage in T-cell cultures was obtained specially with PD-1

signaling, as assessed by pS139-H2AX detection (Fig. 2E). Additionally, expression of most of the tested Th1, Th2 and Th17 cytokines was downmodulated with PD-1, LAG-3 or PD-1/LAG-3 signaling (Fig. 2F). Overall, these results agreed with dysfunctional T-cell phenotypes.

## Identification of common functional networks regulated by constitutive PD-1, LAG-3 and PD-1/LAG-3 signaling in T-cells

To identify common pathways regulated by PD-1, LAG-3 and PD-1/LAG-3 co-signaling, the proteomes of all engineered T-cell lines were analyzed by quantitative differential proteomics. Three biological replicate cultures were used for each cell line with the exception of PD-1 and PD-1 + LAG-3 (two independent cultures). A total number of 2854 proteins was identified of which 833 ($p \le 0.05$) and 539 ($p \le 0.01$) were differentially expressed (Fig. 3A). The top enriched canonical pathways and biological processes for the differential proteomic dataset were related to RNA metabolism and cell signaling pathways including EIF2 and mTOR pathway, and others listed in (Fig. 3B,C). Overall, the associated interactomes corresponded to altered downstream TCR signal transduction pathways, including NF-κB signaling, cell growth through mitotic cell cycle checkpoints and cytokine signaling. The top 5 upstream regulators were identified (Fig. 3D), which corresponded to regulators of cell growth, proliferation, gene expression, protein synthesis and response to environmental stress and cytokines. Reactome, Metascape, STRING, and IPA analyses confirmed these observations.

Then, we sought to identify the interactomes selectively regulated by PD-1, LAG-3 and their combination. To achieve this, the proteome of SC3 control cells was used as a comparative standard (Fig. 3E). PD-1 + LAG-3 T-cells showed the most divergent proteomes with the highest number of differential proteins (Figs. 3A,E and EV3A,B). A major part of the regulated proteome was unique for this combination which indicated altered RNA processing, metabolism and intracellular transport.

Then, we studied common potential regulators of PD-1, LAG-3 and PD-1 + LAG-3 signaling, finding 35 differentially regulated proteins (Fig. 3E,F). Common upstream regulators associated to these set of proteins were identified (Fig. 3G). These proteins were involved in functional interactome networks regulating RNA processing, epigenetic regulation, histone modifications, apoptosis, and transcription (Figs. 3G and EV4), including downregulation of the

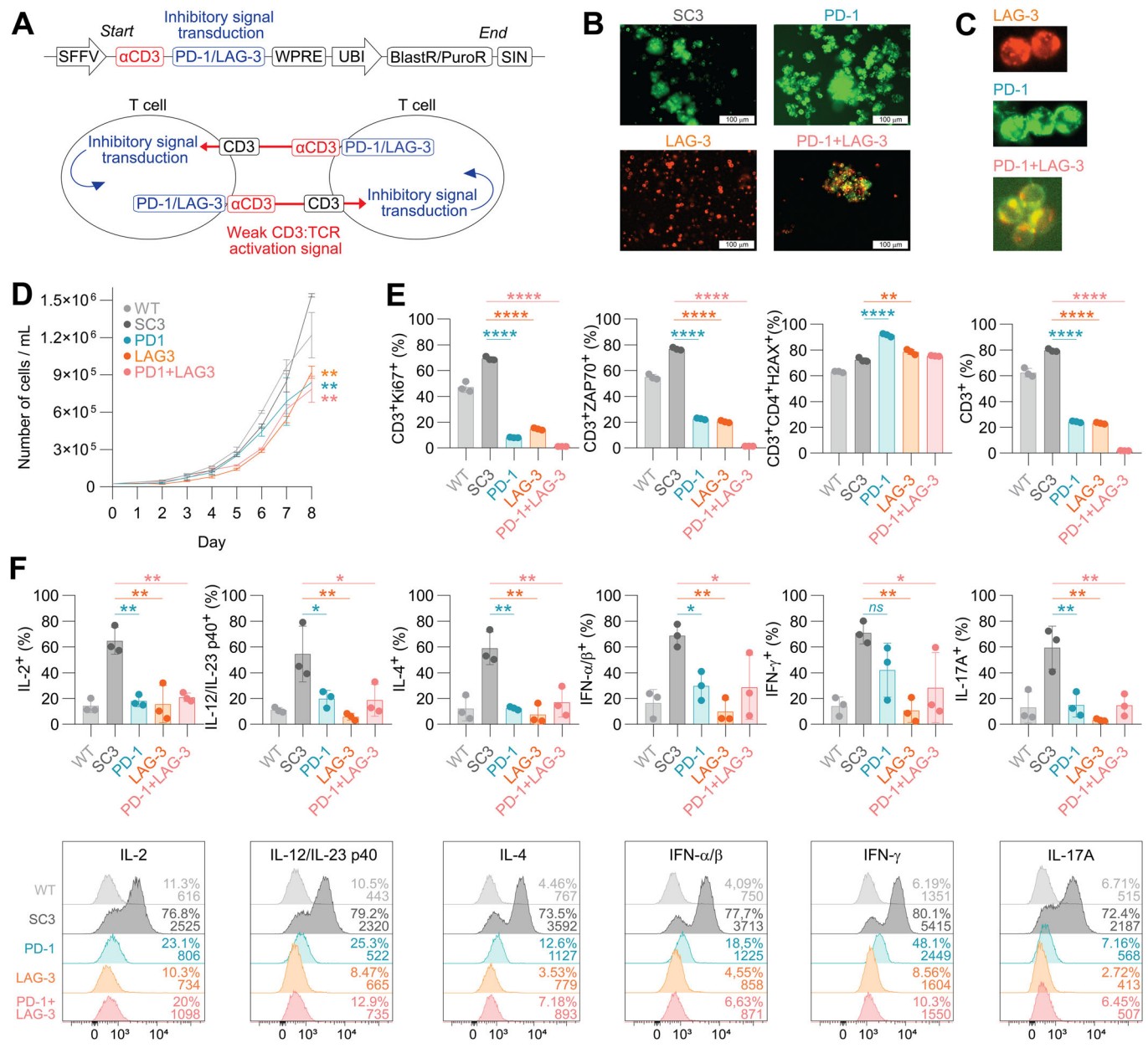

**Figure 2. Expression of molecules with constitutive PD-1 and LAG-3 signaling functions in T cells.**

(A) Top, molecular organization of the two genes made of an anti-CD3 single-chain antibody (SC3) coding sequence fused to LAG-3 or PD-1 stem-transmembrane-intracellular regions. Bottom, mode of action of the engineered molecules expressed by two neighbouring T-cells. The SC3 domain binds and delivers a weak signal to the corresponding TCR through CD3 binding. PD-1 and LAG-3 signaling domains deliver the corresponding inhibitory signals. (B) Fluorescence microscopy pictures of living T-cells expressing SC3-PD-1-GFP (green), SC3-LAG-3-Cherry (red) or both (yellow). (C) Detail of engaging T-cells expressing the constructs. Scale bars and inserts are indicated in the original pictures in the Source Data file for this figure panel. (D) Growth of T-cell lines expressing the indicated constructs ($n = 2$–$5$ independent counts). Relevant statistical comparisons between the cell lines and control cell lines are shown. (E) Bar graphs of T-cell percentages expressing the selected markers within the indicated T-cell lines ($n = 3$ independent repetitions). Error bars correspond to ±SD. (F) Bar graphs of T-cell percentages expressing the selected indicated cytokines within the T-cell lines shown in the graphs ($n = 3$ independent repetitions). Error bars correspond to ±SD. Flow cytometry histograms show a sample of the replicates, where percentage of expression and Mean Fluorescent Intensity values are indicated. Data information: Statistical comparisons are shown in the graph as indicated in Methods. Briefly, for (D–F) statistical comparisons were carried out by a two-way ANOVA followed by pair-wise Tukey tests. Error bars correspond to ±SD. *, **, ***, indicate $P < 0.05$, $P < 0.01$ and $P < 0.001$ differences. ns, non-significant differences. Source data are available online for this figure.

EIF2 signaling pathway (Fig. 3G), which inhibits protein translation, RNA and protein metabolism and TCR signaling (Fig. EV4).

To validate these results in highly dysfunctional T-cells from NSCLC patients, we first chose YY1, one of the common downregulated proteins of the three PD-1, LAG-3 and PD-1 + LAG-3 proteomes (Fig. 3F). YY1 expression in activated T-cells transcriptionally regulates many survival and repair processes (Chandnani et al, 2023; Simmons et al, 2001), and predicted to be a key regulator of

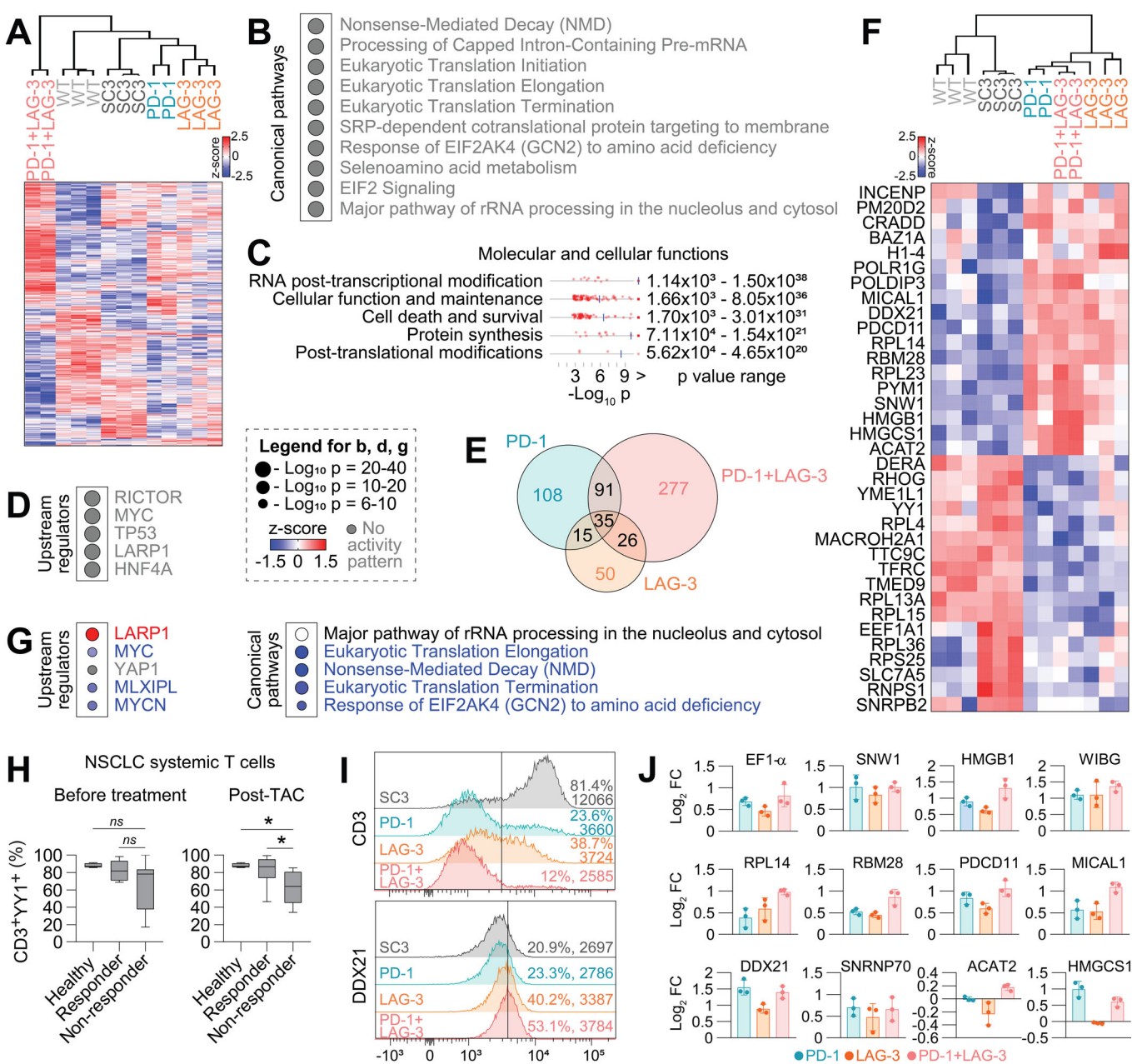

inhibition of T-cell function (Fig. EV4). From the 35 targets, YY1 has been shown to be key in NF-κB, ATF, AP-2α and Myc transcription regulation. Therefore, YY1 was chosen to study its regulation and association with response to immunotherapy (Fig. 3H). In agreement with this, YY1 expression was reduced in T-cells of non-responder NSCLC patients to immunotherapies both before and after the administration of PD-1/PD-L1 blockade antibodies (Fig. 3H). These T-cells are characterized by strong co-expression of PD-1/LAG-3 in contrast to T-cells from responder patients (Edwards et al, 2022; Zuazo et al, 2019). YY1 expression was similar between CD4 and CD8 T cells.

Then, to experimentally demonstrate the down-modulation of the TCR:CD3 proteomic network (Fig. EV4), CD3 and DDX21 expression were evaluated by flow cytometry (Fig. 3I). In agreement with the proteomic profiling, CD3 was strongly downmodulated,

while DDX21 expression was increased by PD-1/LAG-3 co-signaling. To test whether a further selection of these identified targets was transcriptionally regulated, the T-cell lines were transduced with lentivectors expressing GFP under promoters of selected upregulated targets in PD-1/LAG-3 proteomes (Figs. 3F and EV4). Increased transcriptional activity was observed for these promoters compared to control cells represented by fold change in GFP expression (Fig. 3J).

## Identification of a molecular T-cell dysfunctionality programme regulated by PD-1/LAG-3 co-signaling

An analysis of up and downregulated proteins with PD-1, LAG-3 and PD-1 + LAG-3 activated pathway showed similar molecular

**Figure 3.   Proteomes of T-cells with PD-1/LAG-3-regulated molecular pathways.**

(A) Heatmap of regulated proteins from Jurkat T-cell lines with active PD-1, LAG-3 pathways, or their combination, together with unmodified and SC3 cell lines as controls. Heatmap tree structures represent hierarchical clustering based on Euclidean distances. Red, z-score > 0; blue, z-score < 0. $p \leq 0.05$ proteins are plotted. (B) Top 10 enriched canonical pathways associated with the differential proteomic profiles. (C) Top 5 enriched molecular and cellular functions associated with the differential proteomic profiles. (D) Top 5 predicted upstream regulators associated with the differential proteomic profiles. (E) Venn diagram of differentially regulated proteins in the indicated T-cell lines compared to SC3 control cells as a common standard. 35 common proteins were regulated for all conditions. (F) Heatmap of expression of the 35 common proteins as in (E) within the proteomes of all T-cell lines. (G) Column graphs for the top 5 predicted upstream regulators (left) and enriched canonical pathways (right) associated with the 35 commonly regulated proteins. z-scores represent predicted activation (red, z-score > 0) or inhibition (blue, z-score < 0). For white, z-score = 0. (H) Percentage of YY1 expression in healthy donors ($n = 5$), NSCLC anti-PD-1/anti-PD-L1 immunotherapy responders ($n = 10$), and non-responders ($n = 12$) peripheral-blood T cells before treatment and post-TAC timepoints. One-way ANOVA test was used for multi-comparisons, followed by a posteriori Tukey's pair-wise comparisons. Box and whiskers plot with min to max values are plotted, computing the minimum, maximum, median and quartiles. The box extends from the 25th to 75th percentiles. The whiskers go down to the smallest value and up to the largest. (I) Flow cytometry histograms of CD3 and DDXD21 expression in PD-1, LAG-3, and PD-1 + LAG-3 Jurkat T-cell lines. Percentage of expression and Mean Fluorescence Intensity values are indicated. (J) Bar graphs of transcriptional transactivation of the indicated gene promoters identified as regulated by PD-1/LAG-3 signaling in T-cell lines. Promoters expressing GFP were introduced by lentivector transduction in T-cell lines expressing constitutively active PD-1, LAG-3 or PD-1 + LAG-3 molecules as indicated. Transactivation was quantified by GFP expression. Means from 3 independent repetitions are shown, together with standard deviations as error bars. Error bars correspond to ±SD. Data information: Statistical comparisons are shown in the graph as indicated in Methods. Briefly, for (H) one-way ANOVA was used for multi-comparisons, followed by a posteriori Tukey's pair-wise comparisons. *, indicate $P < 0.05$, ns, non-significant differences. For (B, C, D, G), QIAGEN IPA algorithms were used (accessed on 2024), applied on data from curated publicly available datasets of RNA-seq, small RNA-seq, metabolomics, proteomics, microarrays including miRNA and SNP, and small-scale experiments. IPA utilizes two scores for inference; $P$-values from a Fisher's exact test to obtain an enrichment score, and a z-score to assess the match of observed and predicted regulation patterns. Source data are available online for this figure.

functions and protein interaction networks (Figs. 4A and EV3A,B). These included protein translation and elongation, rRNA processing, altered splicing pathways, and downregulation of the EIF2 signaling pathway among other common canonical pathways (Figs. 4A and EV3C). Phenotypic and proteomic data confirmed a reduction of TCR signaling mediators and inhibition of pathways associated to T-cell inactivation (Figs. EV4 and 4A,B), including downmodulation of key components of the TCR signalosome (Fig. EV3C). PD-1/LAG-3 co-signaling showed proteomic profiles associated to pronounced cell growth inhibition with RB1 upregulation and CDK1/CDK6 downregulation, activation of sirtuin signaling with TCR and the NF-κB complex as inhibited upstream regulators (Fig. 4A). Molecular and cellular functions, and diseases and disorders analysis confirmed that T-cells with PD-1/LAG-3 co-signaling showed differential properties compared to T-cells with PD-1 or LAG-3 signaling alone (Fig EV3D,E). Reinforced negative regulation of the TCR signalosome was a major mechanism in PD-1/LAG-3 co-signaling (Fig. 4B). Figure 4B summarises the finding of TCR as an inhibited upstream regulator (in blue) of the differential PD-1/LAG-3 proteomic dataset, explaining the convergence of the observed expression changes (in red and green). Therefore, we performed a detailed analysis of the proteomic dataset in inhibitors of the TCR signalosome and protein translation. This analysis highlighted a specific enrichment in ubiquitin ligases in T-cells with PD-1/LAG-3 co-signaling such as UBA1, RING1, RNF114 and ITCH among others as shown in Fig. 4C. These identified ubiquitin ligases and associated interacting molecules participate in TCR-regulated pathways controlled by CBL E3 ubiquitin ligases (Fig. 4C). Overall, the phenotypic and proteomic data uncovered highly dysfunctional pathways caused by PD-1/LAG-3 co-signaling, including a signature of ubiquitin ligases known to inhibit the TCR signalosome (Figs. EV4 and 4B,C), T-cell activation (Figs. 2 and 4) and cell growth (Figs. 2, 4, and EV3).

## CBL E3 ubiquitin ligases are targeted by PD-1/LAG-3 co-blockade in primary T-cells from lung cancer patients

The proteome profiles showed that PD-1/LAG-3 co-signaling in T-cells was associated to a signature of ubiquitin ligases associated to the CBL E3 pathway that regulate TCR activities. To test if CBL

proteins were expressed by tumor-infiltrating human T-cells, single-cell sequencing data from biopsies of NSCLC patients was analyzed from the single-cell lung cancer extended atlas (LuCA) (Salcher et al, 2022) (Fig. 5A). *CBLB* was brought up among more than 100 ubiquitin-related target genes in lymphoid infiltration, and it was found to be among the top E3 ubiquitin ligases co-expressed with *PDCD1* and *LAG3* in the tumor microenvironment (Dataset EV3). Tumor-infiltrating *PDCD1*+ and *LAG3*+ cells expressed *CBLB*, while a minor proportion expressed *CBLC* (Fig. 5A). Then, we wanted to corroborate these results in peripheral blood T-cells from NSCLC patients resistant to PD-L1/PD-1 blockade immunotherapies, as these T-cells highly co-express PD-1 and LAG-3 (Edwards et al, 2022; Zuazo et al, 2019). These patients showed drastically reduced overall survival (Zuazo et al, 2019) and their T-cells strongly co-upregulated PD-1 and LAG-3 following in vitro stimulation with human lung cancer A549-SC3 stimulator cells (Fig. 5B) as described in (Edwards et al, 2022; Zuazo et al, 2019). The majority of T cells from the NSCLC patient cohort were effector memory and effector T cells as described in (Zuazo et al, 2019). We previously demonstrated in an ex vivo A549 cancer cell-T cell interaction assay that peripheral blood T cells from NSCLC cancer patients need to be TCR stimulated to co-upregulate PD-1 and LAG-3 (Zuazo et al, 2019), and to induce the transcriptional transactivation of E3 ubiquitin ligases (Karwacz et al, 2011). Briefly, human A459 lung adeno-carcinoma cells were engineered to express a membrane bound anti-CD3 single-chain antibody (SC3) to stimulate T cells from NSCLC patients through a 4-day co-culture. The capacities of PD-1/LAG-3 blockers to restimulate these dysfunctional T-cells were assessed by co-culturing them with A549-SC3 stimulator cells. Co-cultures were carried out in the presence of an isotype control, anti-PD-1, anti-LAG-3 and anti-PD-1+anti-LAG-3 antibodies. A bispecific molecule poised to enter clinical development (Humabody CB213) was incorporated in the study (Chocarro et al, 2022a; Edwards et al, 2022). PD-1 blockade inhibited CBL-B upregulation in T-cells from NSCLC patients, as we previously showed in mouse T cells (Fig. 5C). In contrast, LAG-3 blockade alone did not interfere with CBL-B upregulation. Co-blockade with anti-PD-1 and anti-LAG-3 antibodies inhibited CBL-B expression and co-

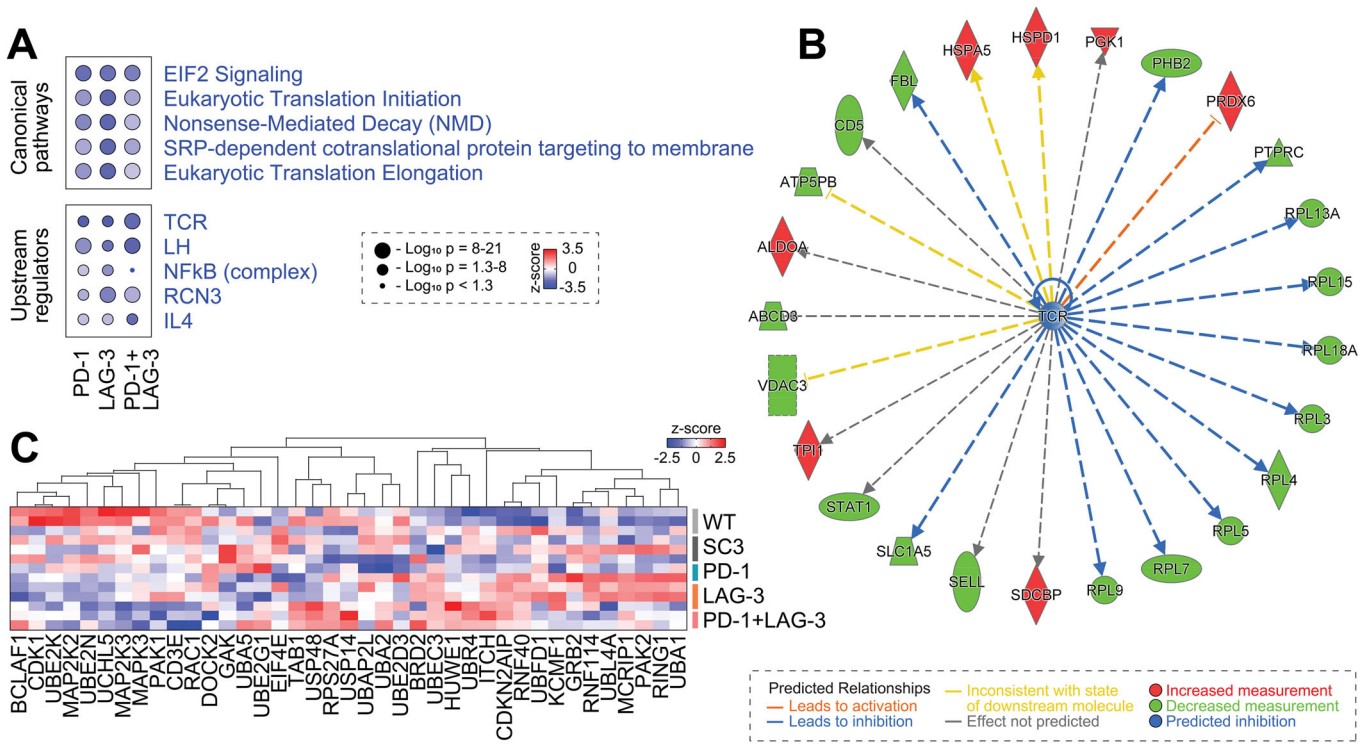

**Figure 4.  Molecular dysfunctionality programme in T-cells by PD-1/LAG-3 co-signaling.**

(A) Enriched canonical pathways and identified upstream regulators. z-scores represent predicted activation (red, z-score > 0), inhibition (blue, z-score < 0). (B) Potential regulation of key functional targets linked to TCR downregulation within the PD-1 + LAG-3 proteome. IPA analysis identified TCR predicted inhibition as an upstream regulator of PD-1, LAG-3 and PD-1 + LAG-3 T-cell proteomes (accessed on 2024). The graph shows relationships between the predicted inhibited TCR (in the centre in blue) and differential expression of the identified proteins in the PD-1 + LAG-3 proteome. In red, upregulated proteins. In green, downregulated proteins. Blue lines, predicted inhibition; orange lines, predicted activation; grey indicates a predicted relationship with a non-predicted effect, and yellow lines, predicted relationship findings inconsistent with the state of the downstream molecule. The specific legends to inter-nodal relationships are described in IPA (Ingenuity Pathway Analysis | QIAGEN Digital Insights). (C) Heatmap of statistically significant differential interactors of the E3 ubiquitin signaling pathway within the indicated T-cell proteomes. Red, significantly upregulated proteins; Blue, significantly downregulated proteins. Heatmap tree structures represent hierarchical clustering based on Euclidean distances. Data information: Statistical comparisons are shown in the graph as indicated in Methods. For (A, B), QIAGEN IPA algorithms were used (accessed on 2024), applied on data from curated publicly available datasets of RNA-seq, small RNA-seq, metabolomics, proteomics, microarrays including miRNA and SNP, and small-scale experiments. IPA utilizes two scores for inference; P-values from a Fisher's exact test to obtain an enrichment score, and a z-score to assess the match of observed and predicted regulation patterns. Source data are available online for this figure.

blockade with CB213 resulted the most potent inhibitor even compared to anti-PD-1 and anti-LAG-3 antibody combination (Fig. 5C). Previous experimental data showed that PD-1 blockade did not inhibit C-CBL expression (Karwacz et al, 2011). In agreement with this, LAG-3 blockade alone, or the anti-PD-1 plus anti-LAG-3 combination did not block C-CBL expression in CD4 and CD8 T-cells from NSCLC patients. In contrast, CB213 significantly blocked C-CBL expression (Fig. 5D). Then, T-cell proliferation was studied by Ki67 expression in stimulated NSCLC T-cells (Fig. 5E). Co-blockade of PD-1 and LAG-3 either with antibodies or with the CB213 bispecific significantly increased proliferation of both CD4 and CD8 T cells. As T-cells recovered proliferative activities, we then evaluated the expression of selected regulator markers of T-cell activation. These markers included SATB1, LCK and SMAD2/3. Bispecific PD-1/LAG-3 co-blockade was tested as it was the most potent T-cell stimulator, leading to SATB1, SMAD2/3 and LCK downregulation, consistent with the modelling of PD-1/LAG-3 co-expression signature (Fig. 5F). We could not evaluate $T_{regs}$ in our study, as these were below detection in peripheral blood from our NSCLC patient cohort.

## PD-1 and LAG-3 co-blockade immunotherapies in combination with CBL-B inhibition exert significant in vivo therapeutic efficacy

We then hypothesized that PD-1/LAG-3 co-blockade immunotherapies could be potentiated in vivo by reinforcing the inhibition of CBL E3 ubiquitin ligases using pharmacological small molecules. To test if this was the case, we evaluated combination treatments for the in vivo treatment of lung adenocarcinoma tumors derived from Lacun3 cells. This experimental model was chosen because it is notoriously refractory to conventional PD-1/PD-L1 blockade immunotherapies (Baraibar et al, 2020; Bleau et al, 2014). To that end, we chose a small molecule inhibitor of CBL-B (CBL-Bi). Importantly, CBL-B inhibitors recently entered clinical development (NCT05662397, NCT05107674). First, we studied whether CBL-Bi inhibited Lacun3 growth in vitro using a wide range of concentrations. No significant cell growth inhibition was achieved at any concentration, indicating that CBL-Bi did not directly affect the growth of Lacun3 cells (Fig. 6A). Then, Lacun3 cells were subcutaneously

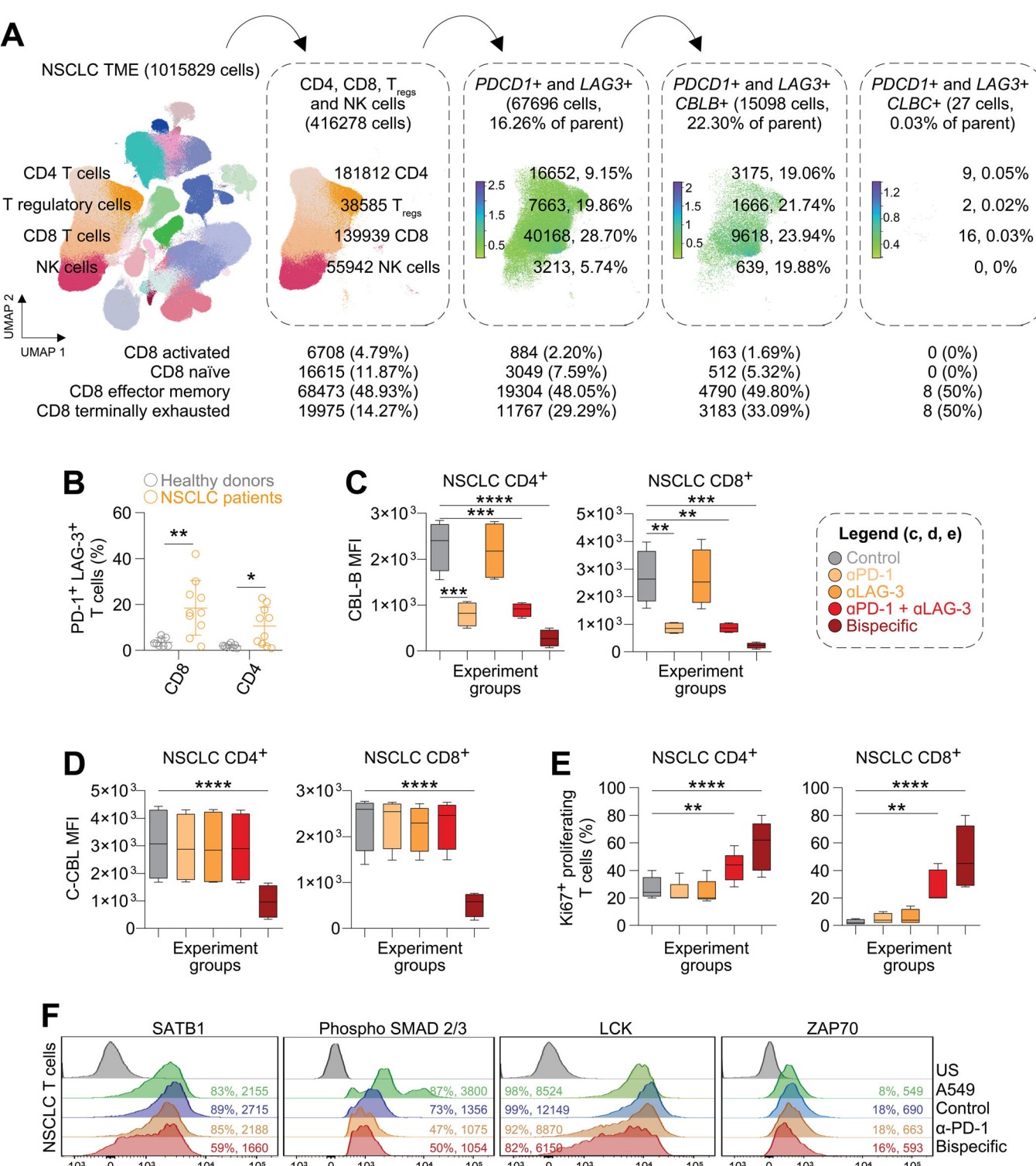

transplanted into groups of mice that were treated with anti-PD-1/anti-LAG-3 antibody combinations with or without CBL-Bi (Fig. 6B,C). The bispecific CB213 could not be used as a treatment as it does not block mouse PD-1 and LAG-3. Tumors were allowed to grow to an average diameter of approximately 3.5 mm before administration of treatments. The growth of Lacun3-derived

tumors was monitored to test for potential anti-tumor effects for any of the treatments for a period of two weeks. A non-significant tendency to reduced tumor growth was observed for all mono-therapies (Fig. 6B,C). Similar results were observed for pair-wise dual combinations. These results indicated that Lacun3-derived tumors were poorly responsive to anti-PD-1, anti-LAG-3 and CBL-

**Figure 5. CBL E3 ubiquitin ligases as targets for PD-1/LAG-3 co-blockade.**

(A) Single-cell sequencing analysis of biopsies from non-small cell lung cancer (NSCLC) patients. Panels indicate the expression of *PDCD1*, *LAG3* and *CBLB* and *CBLC* analyzed from the single-cell lung cancer extended atlas (LuCA) (Salcher et al, 2022) repository as indicated. (B) Dot plot with the percentage of CD4 and CD8 T-cells that co-express PD-1 and LAG-3 after ex vivo activation, from healthy donors (n = 8) and NSCLC patients (n = 10). Statistical comparisons were performed by the Mann–Whitney test. Error bars correspond to ±SD (C) CBL-B expression by mean fluorescent intensities in CD4 and CD8 T-cells from a sample of non-responder NSCLC patients (n = 4), activated ex vivo in the presence of the indicated treatments. Shown data from total CD4 and CD8 gated populations. Statistical comparisons were carried out by a two-way ANOVA to eliminate inter-patient variability followed by pair-wise Tukey tests. Box and whiskers with min to max values are plotted, computing the minimum, maximum, median and quartiles. The box extends from the 25th to 75th percentiles. The whiskers go down to the smallest value and up to the largest. (D) Same as (C) but for C-CBL expression. Box and whiskers with min to max values are plotted, computing the minimum, maximum, median and quartiles. The box extends from the 25th to 75th percentiles. The whiskers go down to the smallest value and up to the largest. (E) Percentage of proliferating CD4 T cells (left) and CD8 T cells (right) from a sample of high PD-1/LAG-3 co-expression patients before starting immunotherapy, activated ex vivo by A549-SC3 cells in the presence of the indicated antibodies. Statistical comparisons were carried out by a two-way ANOVA to eliminate inter-patient variability followed by pair-wise Tukey tests (n = 5). Box and whiskers with min to max values are plotted, computing the minimum, maximum, median and quartiles. The box extends from the 25th to 75th percentiles. The whiskers go down to the smallest value and up to the largest. (F) Flow cytometry histograms of SATB1, Phospho SMAD 2/3, LCK and ZAP70 expression. Gates were established according to unstained controls in T-cells from a sample of non-responder NSCLC patients. Percentage of expression and Mean Fluorescence Intensity values are indicated. Data information: Statistical comparisons are shown in the graph as indicated in Methods. Briefly, for (B) statistical comparisons were performed by the Mann–Whitney test. For (C–E), statistical comparisons were carried out by a two-way ANOVA to eliminate inter-patient variability followed by pair-wise Tukey tests. Error bars correspond to ±SD. **, ***, ****, indicate P < 0.01, P < 0.001 and P < 0.0001 differences. Source data are available online for this figure.

Bi monotherapies. Importantly, only the triple anti-PD-1 + anti-LAG-3 + CBL-Bi combination showed a significant delay in tumor growth (Fig. 6B,C). To find out if this anti-tumor activity translated into a significant increase in survival, we focused on the triple combination but including three different doses of CBL-Bi (10 mg/kg, 20 mg/kg and 30 mg/kg) (Fig. 6D). Importantly, the triple combination with any of the three tested CBL-Bi doses significantly delayed tumor growth at all concentrations, associated to a very highly significant increase in survival (Fig. 6E–G). Importantly, the combination therapy of PD-1/LAG-3 co-blockade with 30 mg/kg of CBL-Bi more than doubled the survival of the saline, and almost duplicated the survival of the dual anti-PD-1/anti-LAG-3 combination (Fig. 6F,G).

To identify the main effector cell type for antitumor efficacy in our model, PD-1/LAG-3/CBL-B were blocked in mice depleted of CD8, CD4 or NK cells (Fig. 7A). Again, a very highly significant increase in survival was achieved for the triple combination, which more than tripled survival of PD-1/LAG-3 co-blockade alone (Fig. 7B). Importantly, only CD8 T cell depletion completely abrogated the anti-tumor efficacy of the triple PD-1/LAG-3/CBL-B blockade combination both in terms of survival (Fig. 7B) and tumor growth (Fig. 7C,D).

## Discussion

Here we carried out thorough and systematic omic studies at multiple levels to identify the regulatory pathways that were activated/deactivated in PD-1/LAG-3+ cancer-associated T cells. This information was used to construct the key regulatory pathways, identifying those that were suitable candidates for pharmacological intervention. The E3 ubiquitin pathway was selected, although our study uncovered several other pathways that could be pharmacologically intervened.

Co-upregulation of LAG-3 with PD-1 is a major mechanism of resistance to conventional PD-L1/PD-1 blockade (Grosso et al, 2009; Johnson et al, 2018; Matsuzaki et al, 2010; Saleh et al, 2019; Woo et al, 2012; Zuazo et al, 2019). This is evidenced by the recent clinical implementation of anti-PD-1 and anti-LAG-3 combinations (Burova et al, 2019; Chocarro et al, 2022b; Ghosh et al, 2019;

Jiang et al, 2021; Sordo-Bahamonde et al, 2021; Tawbi et al, 2022; Zahm et al, 2021). Importantly, although LAG-3 has been extensively described as a key potential immunomodulatory molecule, its mechanism of action remains to be controversial and further studies are required to characterize its signaling in T cells (Aggarwal et al, 2023). Here, we carried out the most extensive and detailed study up-to-date on PD-1/LAG-3 co-expression as a signature of potent immune dysfunctionality in human cancers and T-cells. Overall, it included more than 12000 TCGA samples and established the relationships between *PDCD1/LAG3* gene expression with immune cell infiltration and an extensive number of biomarkers within the tumor-immune infiltrate. A positive correlation between the *PDCD1/LAG3* signature with strong T-cell infiltration and specific pro-inflammatory immune cell types was shown. This demonstrated the association of the *PDCD1/LAG3* signature with potentially immunogenic "hot" cancers. This study encompassed almost 400 genes of T-cell function and activity, and correlations with more than 1000 regulators of cell cycle, cell-to-cell communication, chromatin organization, DNA repair, DNA replication, pro-grammed cell death and autophagy. Overall, these changes indicated pathways associated to immune dysfunctionality and impaired T-cell function, including functions related to the TCR signalosome, cytokine expression, inflammation, cell cycle, immunoproteasome, and signal transduction pathways among others. Thus, a gene expression profile associated to *PDCD1/LAG3* could be extracted from tumors which indicated immune dysfunctionality, possibly from tumor-infiltrating T-cells.

Dysfunctional features indicated by multiomic studies were compared to those in T-cell lines with active PD-1, LAG-3 and PD-1 + LAG-3 signaling pathways. The T-cell phenotyping was consistent with PD-1 and LAG-3 signaling and was accentuated in T-cells with PD-1 + LAG-3 co-signaling. Interestingly, we found only 35 proteins commonly regulated by PD-1, LAG-3 and PD-1 + LAG-3 co-signaling. The engineering of PD-1 + LAG-3 T-cell lines uncovered several T-cell dysfunctions associated with this signature. For example, mTOR-AKT pathways were specifically upregulated and several pathways downmodulated such as the EIF2 pathway that controls translation, transcription, cellular stress responses and intracellular calcium signaling.

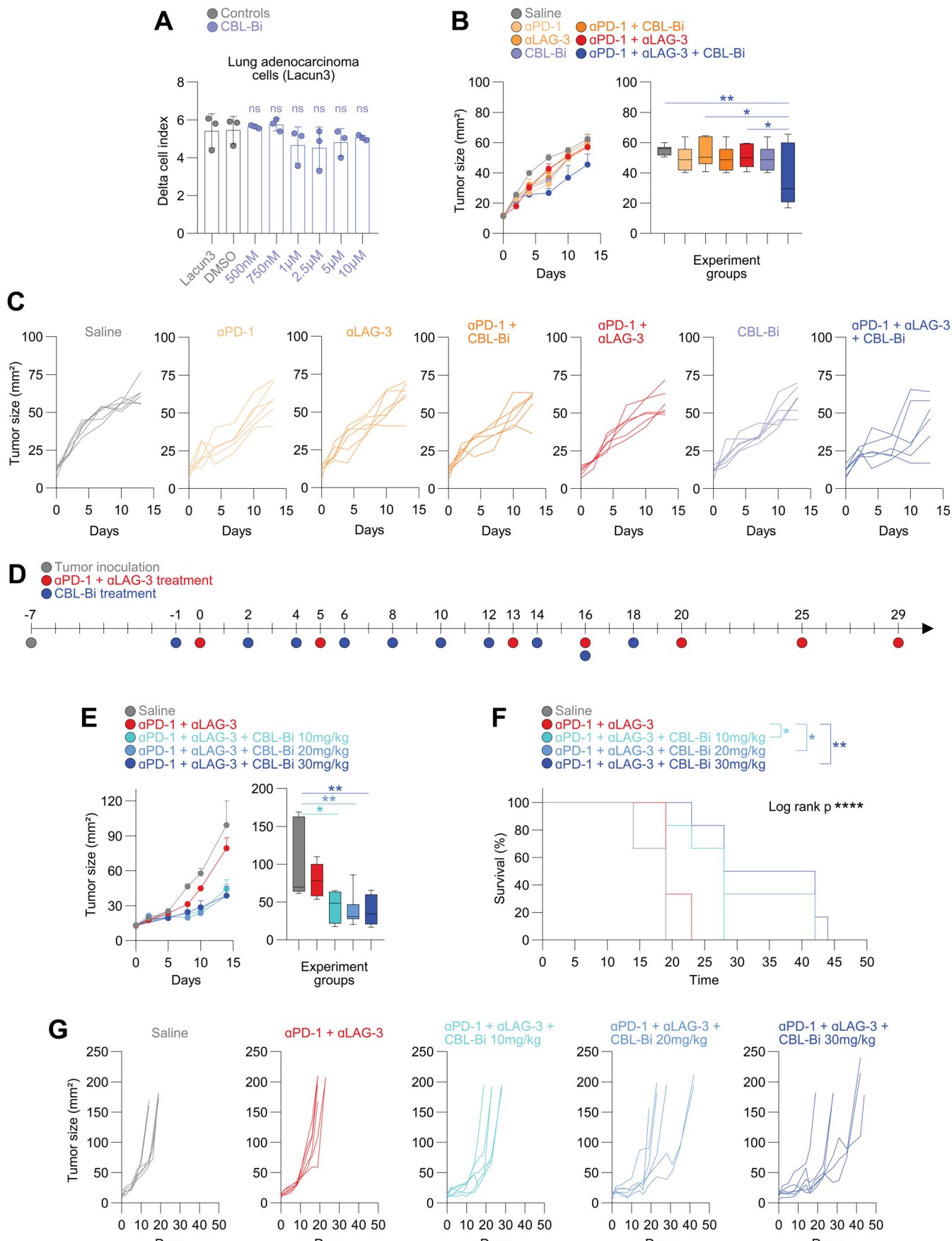

◄

**Figure 6. PD-1, LAG-3 and CBL-B triple blockade has significant therapeutic antitumor effect.**

(A) Real-Time Quantitative Cell Analysis (RTCA) of Lung adenocarcinoma (Lacun3) cells incubated for 70 h with growing concentrations of CBL-B inhibitor (CBL-Bi) as indicated. Error bars correspond to ±SD. Statistical comparisons were carried out by a two-way ANOVA followed by pair-wise Tukey tests ($n = 3$ independent cultures). (B) Mean tumor size following the indicated treatments (left). Tumor volumes 10 days after treatment initiation (right). Error bars correspond to ±SEM (left) and box and whiskers with min to max values (right), computing the minimum, maximum, median and quartiles. The box extends from the 25th to 75th percentiles. The whiskers go down to the smallest value and up to the largest ($n = 6$ mice per group). Briefly, BALB/c female mice were randomly allocated and subcutaneously injected with $2 \times 10^6$ Lung adenocarcinoma (Lacun3) cells per animal. When tumor growth reached an average diameter of 3.5 mm (day 0), 100 μg of anti-PD-1 mAb and 100 μg of anti-LAG3 mAb were administered intraperitoneally (i.p) at days 0, 5 and 13. Control mice received the same volume of saline. Some groups of mice received 30 mg/kg of CBL-bi at days −1, 2, 4, 6, 8, 10 and 12. As negative control, the same volume of saline was injected. Mice were humanely sacrificed at day 14. Statistical comparisons were carried out by a two-way ANOVA followed by pair-wise Tukey tests. (C) Tumor growth of individual mice in the indicated treatment groups ($n = 6$ mice per group). (D) Schematic design of the experiment. BALB/c female mice were randomly allocated and subcutaneously injected with $2 \times 10^6$ Lung adenocarcinoma (Lacun3) cells per animal. When tumor growth reached an average diameter of 3.5 mm (day 0), 100 μg of anti-PD-1 and 100 μg of anti-LAG3 were administered intraperitoneally at days 0, 5, 13, 16, 20, 25 and 29. Control mice received the same volume of saline. Some groups of mice received 10 mg/kg, 20 mg/kg and 30 mg/kg of CBL-Bi at days −1, 2, 4, 6, 8, 10, 12, 14, 16 and 18. As negative control, the same volume of saline was injected. The two perpendicular tumor diameters were measured every two days. Mice were humanely sacrificed when tumor size reached ~150–200 mm$^2$, or when tumor ulceration or discomfort were observed. (E) Evolution of mean tumor size following the indicated treatments (left). Tumor volumes 14 days after treatment initiation (right). Error bars correspond to ±SEM (left) and box and whiskers with min to max values (right), computing the minimum, maximum, median and quartiles. The box extends from the 25th to 75th percentiles. The whiskers go down to the smallest value and up to the largest ($n = 6$ mice per group). Statistical comparisons were carried out by a two-way ANOVA followed by pair-wise Tukey tests. (F) Kaplan–Meier survival plot of mice under the indicated treatments (percent). Statistical significance was tested with the Log-rank test. (G) Tumor growth of individual mice in the indicated treatment groups ($n = 6$ mice per group). Data information: Statistical comparisons are shown in the graph as indicated in Methods. Briefly, for (A, B, E), statistical comparisons were carried out by a two-way ANOVA followed by pair-wise Tukey tests. For (F), Survival was represented by Kaplan–Meier plots and analyzed by log-rank test. *, **, ****, indicate $P < 0.05$, $P < 0.01$ and $P < 0.0001$ differences. Source data are available online for this figure.

PD-1 plays key roles in signaling at different levels (Hui et al, 2017; Kamphorst et al, 2017; Karwacz et al, 2011. EMBO Mol Med). First, during antigen presentation by professional antigen presenting cells to T cells, that requires in most instances CD28-CD80 costimulation. Then, at the tumor site, where CD28-CD80 interactions are in most instances absent between cancer cells and T cells, as effector T cells only require TCR stimulation through peptide-MHC binding to exert cytotoxicity. In this context PD1-PDL1 interactions would interfere with TCR signaling in the absence of CD28 interactions. It needs to be remarked that the assays utilized here do not include CD28 signaling. Our data may better reflect the mechanisms taking place within the tumor environment but not during antigen presentation by professional antigen presenting cells, where PD-1-PD-L1 signaling plays a major regulatory role as shown before (Karwacz et al, 2011).

The accentuated negative regulation of the TCR signalosome was central in the PD-1/LAG-3 dysfunctional signature. PD-1/LAG-3 T-cells were found to be enriched in negative regulators of the TCR signalosome components and protein translation, particularly members of E3 ubiquitin ligase pathways (Bachmaier et al, 2000; Clemens, 2001; Chiang et al, 2000; Dikic et al, 2003; Karwacz et al, 2011; Shamim et al, 2007). Our results highlighted a signature associated with E3 ubiquitin ligase pathways in our proteome. However, the coverage of the proteomic database in this study did not include its master regulator CBL-B, although we could detect its expression in both our Jurkat T cell lines and in primary T cells stimulated with cancer cells. As the relationship between the tumor and immune cells is dynamic and phenotypes change with time, it could be possible that CBL-B negative T cells may also have an impact on ICI resistance. As a CBL-B inhibitor is currently under clinical evaluation alone, in combination with chemotherapy (clinicaltrials.gov ID: NCT05107674) and in combination with anti-PD-1 antibodies (clinicaltrials.gov ID: NCT05662397), we chose CBL-B as a therapeutic target in combination with PD-1/LAG-3 co-blockade. Thus, we found that CBL-B and C-CBL were differentially regulated by PD-1/LAG-3 by co-blockade studies in primary T-cells from NSCLC patients.

Notably, ubiquitination via CBL-B and proteasomal degradation may play a major role not only on PD-1/LAG-3 signaling but also on the post-transcriptional regulation of other immune checkpoints such as CD226, TIGIT or CD28-related signaling molecules (Braun et al, 2020). Our results identified CBL E3 ubiquitin ligases as potential druggable targets that could be combined with conventional PD-1 and LAG-3 co-blockade. Indeed, notable antitumor activities were demonstrated in vivo in a lung cancer model refractory to immunotherapy. The triple anti-PD-1/anti-LAG-3/CBL-Bi combination more than tripled survival, providing a strategy to improve current immunotherapy strategies. Our results may provide an effective approach to overcome PD-1/LAG-3-mediated resistance by blocking CBL E3 ubiquitin ligases, which could have a significant clinical impact.

## Methods

### Patients and clinical samples

Blood samples were prospectively collected from a cohort of patients with locally advanced or metastatic NSCLC treated with IC inhibitors (ICIs, nivolumab, pembrolizumab, atezolizumab) following current guidelines (Herbst et al, 2016; Horn et al, 2017; Rittmeyer et al, 2017) at University Hospital of Navarre. The prospective observational study was approved by the Ethics Committee of Clinical Investigations at the University Hospital of Navarre (reference number: PI_2020/115). Informed consent was obtained from all subjects, and all experiments conformed to the principles in the WMA Declaration of Helsinki and the Department of Health and Human Services Belmont Report. Clinical details and composition of the cohort under study are described elsewhere (Arasanz et al, 2020; Bocanegra et al, 2023; Zuazo et al, 2019). Briefly, eligible patients were 18 years of age or older who had progressed to first-line platinum-based chemotherapy or concurrent chemoradiotherapy. A CT scan before the beginning of immunotherapy and another one after the first cycle of

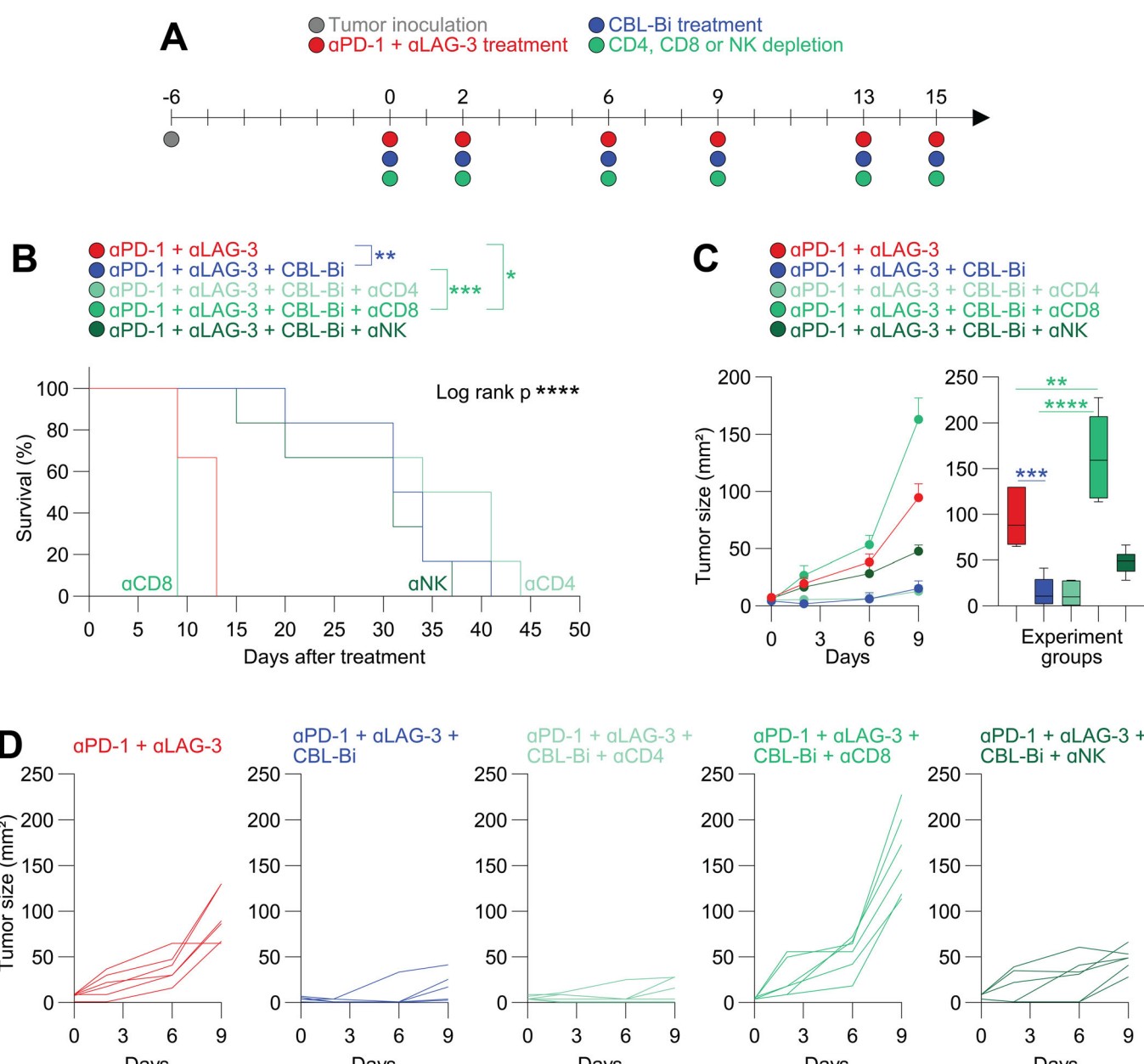

**Figure 7. CD8 T cells are responsible for the PD-1, LAG-3 and CBL-B triple blockade significant therapeutic antitumor effect.**

(A) Schematic design of the experiment. BALB/c female mice were randomly allocated and subcutaneously injected with $2 \times 10^6$ Lung adenocarcinoma (Lacun3) cells per animal. 100 μg of anti-PD-1, 100 μg of anti-LAG3, 30 mg/kg of CBL-Bi and the corresponding depletion antibodies were administered intraperitoneally at days 0, 2, 6, 9, 13 and 15 as indicated in the figure. NK, CD4, and CD8 T-cell depletions were carried out by intraperitoneal administration of 100 μg of anti-mouse CD8a, CD4 or NK1.1 antibody. Mice were humanely sacrificed when tumor size reached ~150–200 mm², or when tumor ulceration or discomfort were observed. (B) Kaplan–Meier survival plot of mice under the indicated treatments or depletion (percent). Statistical significance was tested with the Log-rank test. (C) Evolution of mean tumor size following the indicated treatments (left). Tumor volumes 9 days after treatment initiation (right). Error bars correspond to ±SEM (left) and box and whiskers with min to max values (right), computing the minimum, maximum, median and quartiles for 25th and 75th percentiles. The whiskers go down to the smallest value and up to the largest ($n = 6$ mice per group). Data information: Statistical comparisons were carried out by a two-way ANOVA followed by pair-wise Tukey tests. (D) Tumor growth of individual mice in the indicated treatment groups ($n = 6$ mice per group). Statistical comparisons are shown in the graph as indicated in Methods. Data information: Briefly, for (B), survival was represented by Kaplan–Meier plots and analyzed by log-rank test. For (C), statistical comparisons were carried out by a two-way ANOVA followed by pair-wise Tukey tests. *, **, ****, indicate $P < 0.05$, $P < 0.01$, and $P < 0.0001$ differences. Source data are available online for this figure.

immunotherapy were performed. Blood samples from NSCLC patients were collected prior to treatment and before administration of each immunotherapy cycle.

Responses were assessed following standard protocols according to current clinical practice based on RECIST 1.1 (Eisenhauer et al, 2009) and Immune-Related Response Criteria (Wolchok et al, 2009).

## Cells and lentivectors

Jurkat T cells were purchased from the ATCC, and cultured in standard conditions in RPMI medium supplemented with 10% FCS, penicillin/streptomycin and glutamine. Jurkat T cell lines were obtained by lentivector transduction and selection with puromycin, blasticidin or both in complete RPMI medium. Human embryonic kidney (HEK) 293T cells were purchased from the ATCC and grown in DMEM supplemented with 10% FCS, penicillin/streptomycin and glutamine. Cells were grown following standard procedures. Cells were authenticated by the ATCC and verified to be mycoplasma-free.

Patient or healthy donor peripheral blood mononuclear cells (PBMCs) were isolated by FICOL gradients immediately after the blood extraction. PBMCs were washed and T cells were isolated as described (Zuazo et al, 2019). T cells were maintained in TexMACs medium (Miltenyi) until use.

The engineering and culture of SC3-A549 cells are described elsewhere (Zuazo et al, 2019). Briefly, these cells are human lung adenocarcinoma cells modified to express an anti-CD3 single chain antibody to stimulate T cells. Co-cultures of SC3-A549 cells with primary T cells were carried out as described at a 2:1 ratio (Zuazo et al, 2019). These cell cultures were performed in the presence of immune checkpoint inhibitors. PD-1, LAG-3, control isotype antibodies and their working concentrations used for the assays are described in (Zuazo et al, 2019). The bispecific Humabody CB213 that blocks simultaneously PD-1 and LAG-3 together with its control and their working concentrations are described in (Edwards et al, 2022) Co-cultures were carried out for two days before analyses.

## Plasmids and lentivector production

The coding sequences for LAG-3 and PD-1 genes were retrieved from https://www.ncbi.nlm.nih.gov/gene and https://www.uniprot.org/. Their cDNA sequences were synthetized (Geneart, Thermofisher) flanked by BamHI and NotI sites. Their extracellular stem, transmembrane and intracellular domains were amplified by PCR and fused to the extracellular domain of the SC3 construct in lentivectors as described before (Zuazo et al, 2019). Briefly, this molecule is a single-chain antibody derived from the anti-CD3 antibody OKT3 which is fused in frame to a human IgG transmembrane domain cloned under the transcriptional control of the SFFV promoter in a pDUAL-lentivector. For the constructs containing GFP or mCherry, their coding sequences were fused in frame to the intracellular signaling domains of PD-1 or LAG-3. All constructs were sequenced.

All the fusion genes were cloned into pDUAL-PuroR or pDUAL-BlastR lentivectors under the transcriptional control of the SFFV promoter. pDUAL-PuroR or pDUAL-BlastR lentivectors express puromycin or blasticidin resistance under the transcriptional control of the human ubiquitin promoter, and these are

described in (Gato-Canas et al, 2017). To engineer lentivectors containing T-cell specific promoters, the promoter sequences for the following genes were retrieved from EPD Eukaryotic Promoter Database (http://epd.vital-it.ch/human/human_database.php): *EF-1A, SNW1, HMGB1, WIBG, RPL14, RBM28, PDCD11, MICAL1, DDX21, SNRP70, ACAT2, HMGCS1*. Two kilobases upstream of the transcription start site were synthetized for each promoter (Geneart, Thermofisher), flanked by EcoRI and AscI restriction sites. Then, the SFFV promoter from the pSIN-GFP lentivector (Escors et al, 2008) was replaced by each of the T-cell-specific promoters.

Lentivector production and titration were carried out as described elsewhere (Karwacz et al, 2011; Liechtenstein et al, 2014; Selden et al, 2007). Jurkat T-cells were transduced with a multiplicity of 10 and selected with puromycin (Gibco), blasticidin (Gibco) or both as described (Gato-Canas et al, 2017). Transduced cells were analyzed for the expression of the target of interest by flow cytometry or by fluorescence microscopy. BioTeck Citation™ 5 cell imaging multi-mode reader-based fluorescence detection was used.

## Flow cytometry

Surface and intracellular stainings were carried out as described (Zuazo et al, 2019). The following antibodies were used at 1:50 dilution unless otherwise stated: CD4-APC-Vio770 (clone M-T466, Miltenyi), CD3-APC (clone REA613, Milenyi Biotec), CD28-PECy7 (clone CD28.2, Biolegend), PD-1-PE (clone EH12.2H7, Biolegend), CD8-FITC (clone SDK1, Biolegend), LAG3-PE (clone 11C3C65, Biolegend), LAG3-PerCP-Cy5.5 (clone 11C3C65, Biolegend), CD4-FITC (clone REA623, Milteny), CD3-PerCP-Cy5.5 (clone T100, TONBO), CD4-PE-Vio770 (clone REA261, Milteny), CD223-PerCP-Cy5 (clone 11C3C65, Biolegend), PD-1-Pacific Blue (clone EH12.2H7, Biolegend). When required, surface staining was followed by intracellular staining using the BD Transcription Factor buffer set or the BD Fixation/permeabilization Kit. The following antibodies were used for intracellular staining at 1:100 dilution unless otherwise stated: anti-Ki67-APC or pacific blue-conjugated antibodies (clone ki67, Biolegend, dilution 1:10), AF648-CBL-B (clone G1, Santa Cruz Biotech, dilution 1:50), H2AX-FITC (clone 2F3, Biolegend), AF648-C-CBL (clone A-9, Santa Cruz Biotech, dilution 1:50), ZAP-70-PE (A16043B, Biolegend, dilution 1:10 or 1:100), IFN-γ-FITC (clone REA600, Milteny), IFN-α/β-PE (clone FAB245P, RD Systems), IL-12/IL-23 p40-AF647 (clone C11.5, Biolegend), IL4-PE (clone REA895, Milteny), IL2-APC (clone 17H12, Biolegend), IL17A-BV421 (clone BL168, Biolegend), AF647 anti-SATB1 (clone 14/SATB1, BD Biosciences, dilution 1:10), anti-Smad2 (pS465/pS467)/Smad3 (pS423/pS425)-PE (clone 072-670, BD Biosciences, dilution 1:10), AF647 anti-Lck (pY505) (clone 4/LCK-Y505, BD Biosciences, dilution 1:10), YY1 (clone EPR4652, Abcam, dilution 1:500) and anti-rabbit IgG AF488 conjugate (clone polyclonal, Invitrogen, dilution 1:200) as secondary antibody. Flow cytometry was carried out with a BD FACS CANTO flow cytometer. Data was analyzed by Flowjo.

## Mouse tumor models and therapies

Animal studies were approved by the University of Navarra ethics committee (E20-22(078-19E1) and 077-19) and from the

Government of Navarra. All animals were housed at CIMA's animal house facilities (conventional biosafety 2 housing conditions with environmental enrichment, ES31 2010000132, University of Navarre). ARRIVE reporting guidelines were followed. Sample sizes were calculated to achieve a minimum power of 0.8 for F-based tests taking into consideration a large effect size ($f = 04$). Power calculations were carried out with Gpower 3.1.9.7. Three independent in vivo experiments were carried out. For all, BALB/c female mice of 6 weeks of age were randomly allocated and subcutaneously (s.c) injected with $2 \times 10^6$ Lacun3 cells per animal. Lacun3 cells were a kind gift from Prof Luis Montuenga (Cima-Universidad de Navarra). No blinding was established for the experiments. Anti-tumor T-cell responses take a minimum of 10 days to act (Karwacz et al, 2011). Therefore, tumors were allowed to grow up to an average diameter of 3.5 mm (day 0), before starting immunotherapies, as the growth of Lacun3 tumors show fast kinetics. Then, 100 µg of anti-PD-1 mAb (RPMI-14, BioXCell), 100 µg of anti-LAG3 mAb (C9B7W, Abyntek) were administered intraperitoneally (i.p) following the scheme described in the figures. Antibodies were administered intraperitoneally following standard procedures in mouse immunotherapy models. Control mice received the same volume of saline. Some groups of mice received CBL-b inhibitor (Cbl-b-IN-3, HY-141432) at the times indicated in the figure and at 10 mg/kg, 20 mg/kg and 30 mg/kg of CBL-b inhibitor (Cbl-b-IN-3, HY-141432) where indicated. As negative control, the same volume of saline was injected. In the first in vivo experiment, mice were humanely sacrificed at day 14 for evaluation. The two perpendicular tumor diameters were measured every 2 days. The size was calculated using the formula: Size = Length × Width. In the second in vivo experiment, mice were humanely sacrificed when tumor size reached ~150–200 mm², or when tumor ulceration or discomfort were observed. Blinding was used for data analysis and correlation with survival. For in vivo depletion experiments, CD4, CD8 or NK depletions were carried out by intraperitoneal administration of 100 µg of anti-mouse CD8a (clone 2.43; BioXCell), CD4 (clone GK1.5; BioXCell) or NK1.1 (clone PK136; BioXCell) antibodies following routine techniques in our group (Blanco et al, 2024; Bocanegra et al, 2023). Mice were humanely sacrificed when tumor size reached ~150–200 mm², or when tumor ulceration or discomfort were observed.

## Real-time cell analysis (RTCA)

Cytotoxicity in cell cultures was evaluated by xCELLigence Real-Time Cell Analysis (RTCA) (Roche Diagnostics GmbH, Mannheim, Germany) as described before (Blanco et al, 2024; Bocanegra et al, 2023; Gato-Canas et al, 2015). Briefly, Lacun3 cells were seeded at a density of $3 \times 10^3$ cells/well on gold microelectrode-embedded 16-well microplates (E-plates; Roche Diagnostics, Basel, Switzerland) and incubated at 37 °C with 5% $CO_2$. Impedance was recorded at 15 min intervals. CBL-B inhibitor CBL-b inhibitor (Cbl-b-IN-3, HY-141432) was added to the culture at seeding time at 500 nM, 750 nM, 1 µM, 2.5 µM, 5 µM and 10 µM as indicated in Fig. 6A. DMSO was added at seeding time as a negative control. All incubations were performed in a volume of 100 µl. 70 h Delta Cell Index values were evaluated by the RTCA-DP software (Roche Diagnostics GmbH). The Delta CI (Delta Cell index) was used to normalise data.

## Proteomics and data analysis

Cell proteomes were analyzed by SWATH-MS (Collins et al, 2017). Cell pellets were homogenized in a lysis buffer containing 7 M urea, 2 M thiourea, and 50 mM DTT. The homogenates were spun down at $100,000 \times g$ for 1 h at 15 °C. Protein quantitation was performed by Bradford (Bio-Rad). Protein in-solution digestion, peptide purification, and reconstitution prior to mass spectrometric analysis and library generation were performed as previously reported (Ferrer et al, 2021).

Peptides recovered from in-gel digestion processing were reconstituted into a final concentration of 0.5 µg/µL of 2% ACN, 0.5% FA, 97.5% Milli-Q-water prior to mass spectrometric analysis. MS/MS datasets for spectral library generation were acquired on a Triple TOF 5600+ mass spectrometer (Sciex, Canada) interfaced to an Eksigent nanoLC ultra 2D pump system (SCIEX, Canada) fitted with a 75 µm ID column (Thermo Scientific 0.075 × 250 mm, particle size 3 µm and pore size 100 Å). Prior to separation, the peptides were concentrated on a C18 precolumn (Thermo Scientific 0.1 × 50 mm, particle size 5 µm and pore size 100 Å). Mobile phases were 100% water 0.1% formic acid (FA) (buffer A) and 100% Acetonitrile 0.1% FA (buffer B). Column gradient was developed in a gradient from 2% B to 40% B in 120 min. Column was equilibrated in 95% B for 10 min and 2% B for 10 min. During all processes, the precolumn was in line with column and flow was maintained all along the gradient at 300 nL/min. Output of the separation column was directly coupled to nanoelectrospray source. MS1 spectra was collected in the range of 350–1250 $m/z$ for 250 ms. The 35 most intense precursors with charge states of 2 to 5 that exceeded 150 counts per second were selected for fragmentation, rolling collision energy was used for fragmentation, and MS2 spectra were collected in the range of 230–1500 $m/z$ for 100 ms. The precursor ions were dynamically excluded from reselection for 15 s. MS/MS data acquisition was performed using AnalystTF 1.7 (Sciex) and spectra files were processed through ProteinPilot v5.0 search engine (Sciex) using Paragon Algorithm (v.4.0.0.0) (Shilov et al, 2007) for database search. To avoid using the same spectral evidence in more than one protein, the identified proteins were grouped based on MS/MS spectra by the Progroup algorithm, regardless of the peptide sequence assigned. The protein within each group that could explain more spectral data with confidence was depicted as the primary protein of the group. False discovery rate was performed using a nonlinear fitting method (Tang et al, 2008) and displayed results were those reporting a 1% Global false discovery rate or better.

For SWATH-MS-based experiments, the instrument (Sciex Triple-TOF 5600+) was configured as described elsewhere (Gillet et al, 2012). Briefly, the mass spectrometer was operated using an isolation width of 16 Da (15 Da of optimal ion transmission efficiency and 1 Da for the window overlap), a set of 37 overlapping windows were constructed covering the mass range 450–1000 Da. In this way, 1 µL of ach sample was loaded onto a trap column (Thermo Scientific 0.1 × 50 mm, particle size 5 µm and pore size 100 Å) and desalted with 0.1% TFA at 3 µL/min during 10 min. The peptides were loaded onto an analytical column (Thermo Scientific 0.075 × 250 mm, particle size 3 µm and pore size 100 Å) equilibrated in 2% acetonitrile 0.1% FA. Peptide elution was carried out with a linear gradient of 2 to 40% B in 120 min (mobile phases A:100% water 0.1% formic acid (FA) and B: 100%

**The paper explained**

**Problem**

A significant number of cancer patients do not benefit from PD-L1/PD-1 blockade immunotherapies. PD-1 and LAG-3 co-upregulation in T-cells is one of the major mechanisms of resistance by establishing a highly dysfunctional state in T-cells. However, how PD-1 and LAG-3 cooperate to establish T-cell dysfunctionality, the combined effects of their co-signaling, and the molecular effects of co-blockade over PD-1/LAG-3-associated dysfunctional signatures are largely unknown.

**Results**

PD-1/LAG-3 gene co-expression signatures were extracted from multiomic data associated to T-cell functions in human cancers, uncovering a PD-1/LAG-3 highly dysfunctional signature. PD-1 and LAG-3 signaling pathways were co-activated in T-cells and analyzed through quantitative differential proteomics. CBL E3 ubiquitin ligases were found as key target associated to the regulation of central T-cell dysfunctional pathways in T-cell lines and in primary T cells from non-small cell lung cancer (NSCLC) patients. Pharmacologic inhibition of CBL-B combined with PD-1/LAG-3 co-blockade demonstrated notable therapeutic efficacies in a lung cancer model poorly responsive to immunotherapies.

**Impact**

Molecular pathways associated to PD-1 and LAG-3 co-signaling are identified, which mediate resistance to PD-1 monoblockade. A druggable target, CBL-B was identified by genomic and proteomic techniques in T-cells with PD-1/LAG-3 co-signaling. CBL-B inhibition combined with PD-1 and LAG-3 antibody-co-blockade more than tripled survival in mice with lung cancer poorly responsive to immunotherapies. Patients with dysfunctional T-cell immunity resistant to conventional antibody blockade immunotherapies could benefit from immunotherapy blockade combinations with CBL-B inhibitors.

Acetonitrile 0.1% FA) at a flow rate of 300 nL/min. Eluted peptides were infused in the mass spectrometer. The Triple-TOF was operated in swath mode, in which a 0.050 s TOF MS scan from 350 to 1250 $m/z$ was performed, followed by 0.080 s product ion scans from 230 to 1800 $m/z$ on the 37 defined windows (3.05 s/cycle). Collision energy was set to optimum energy for a $2+$ ion at the centre of each SWATH block with a 15 eV collision energy spread. The mass spectrometer was always operated in high sensitivity mode. The resulting ProteinPilot group file from library generation was loaded into PeakView (v2.1, Sciex) and peaks from SWATH runs were extracted with a peptide confidence threshold of 99% confidence (Unused Score ≥1.3) and a false discovery rate (FDR) lower than 1%. For this, the MS/MS spectra of the assigned peptides was extracted by ProteinPilot, and only the proteins that fulfilled the following criteria were validated: (1) peptide mass tolerance lower than 10 ppm, (2) 99% of confidence level in peptide identification, and (3) complete b/y ions series found in the MS/MS spectrum. Only proteins quantified with at least two unique peptides were considered. The quantitative data obtained by PeakView were analyzed using Perseus software (Tyanova et al, 2016) for statistical analysis and data visualization. Output files with the identified proteins were then managed with Perseus for subsequent statistical analysis. Multiple-samples tests were performed. Unpaired Student t-tests were used for direct comparisons between two groups of samples. Differential PD-1/LAG-3 proteins versus the SC3 control condition comparisons were identified,

following $p$-value ≤ 0.05, Log$_2$ (Fold Change) ≥ 0.38 and Log$_2$ (Fold Change) ≤ −0.38 criteria.

## Bioinformatics, computational and statistical analyses

Public transcriptomic and genomic databases for TCGA human cancers were analyzed with TIMER 2.0 (Li et al, 2016; Li et al, 2017; Li et al, 2020), GEPIA (Gene Expression Profiling Interactive Analysis) (Tang et al, 2019; Tang et al, 2017), cBioportal (Cerami et al, 2012; Gao et al, 2013), TNM Plot (Bartha and Gyorffy, 2021), and Ingenuity Pathway Analysis (IPA) (IPA was accessed on 2024) (Kramer et al, 2014).

Differential protein expression data from proteomic analyses by Perseus were analyzed with Ingenuity Pathway Analysis (IPA) (IPA was accessed on 2024) (Kramer et al, 2014), Metascape (Zhou et al, 2019), Reactome (Croft et al), STRING (Szklarczyk et al, 2019). IPA compares a dataset of genes to the Ingenuity Knowledge Base and applies four causal analytic algorithms to identify upstream regulators, mechanistic networks, causal network analysis and downstream effects analysis. IPA utilizes two scores for inference; $P$-values from a Fisher's exact test to obtain an enrichment score, and a Z-score to assess the match of observed and predicted regulation patterns.

Statistical analyses were performed with the GraphPad Prism 8.0.1 software package. No data was discarded from analyses. Normality was evaluated with Kolmogorov–Smirnov and Shapiro-Wilk tests (in the case of samples with $n < 10$). Homocedasticity was evaluated by the chi-squared test. Statistical analyses were performed with paired-tailed Student's t-test (paired dependent t-test), with a significance level of $p < 0.05$. For non-normally distributed data or with intrinsic variability the Wilcoxon matched pairs test was used. Two-way ANOVA test was used for multi-comparisons, followed by a posteriori Tukey's pair-wise comparisons.

Tumor measurements of in vivo experiments were represented as tumor surface (mm$^2$) and plotted either as individual data points for one individual mouse, or as mean ± SEM or as indicated in figure legends. For in vivo experiments, two-way ANOVA test was used for multi-comparisons, followed by a posteriori Tukey's pair-wise comparisons. Survival was represented by Kaplan–Meier plots and analyzed by log-rank test.

## Data availability

Mass-spectrometry data and search results files were deposited in the Proteome Xchange Consortium via the JPOST partner repository: https://proteomecentral.proteomexchange.org/cgi/GetDataset?ID=PXD040408 https://repository.jpostdb.org/entry/JPST002062.

The source data of this paper are collected in the following database record: biostudies:S-SCDT-10_1038-S44321-024-00098-y.

## Peer review information

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

## Acknowledgements

We sincerely acknowledge all the patients and families that generously agreed to participate in this study. We are also thankful to the nursing staff of the Medical Oncology Day Care at University Hospital of Navarre who kindly provided the clinical samples. This research was supported by: The Spanish Association against Cancer (AECC), PROYE16001ESCO; Instituto de Salud Carlos III (ISCIII)-FEDER Project grants FIS PI20/00010, FIS PI23/00196, COV20/00237, and TRANSPOCART ICI19/00069; Biomedicine Project Grant from the Department of Health of the Government of Navarre-FEDER funds (BMED 050-2019, 51-2021, 036-2023); Strategic projects from the Department of Industry, Government of Navarre (AGATA, Ref. 0011-1411-2020-000013; LINTERNA, Ref. 0011-1411-2020-000033; DESCARTHES, 0011-1411-2019-000058); European Union Horizon 2020 ISOLDA project, under grant agreement ID: 848166. LC is financed by Instituto de Salud Carlos III (ISCIII), co-financed by FEDER funds, "Contratos PFIS: contratos predoctorales de formación en investigación en salud" (FI21/00080); ME is financed by the Navarrabiomed-Fundación Miguel Servet predoctoral contract.

## Author contributions

**Luisa Chocarro**: Conceptualization; Data curation; Software; Formal analysis; Validation; Investigation; Visualization; Methodology; Writing—original draft; Writing—review and editing. **Ester Blanco**: Formal analysis; Validation; Investigation; Visualization; Methodology; Writing—original draft. **Leticia Fernandez-Rubio**: Formal analysis; Validation; Investigation; Methodology. **Maider Garnica**: Investigation; Methodology. **Miren Zuazo**: Investigation; Methodology. **Maria Jesus Garcia**: Investigation; Methodology. **Ana Bocanegra**: Investigation; Methodology. **Miriam Echaide**: Investigation; Methodology. **Colette Johnston**: Investigation; Methodology. **Carolyn J Edwards**: Resources; Investigation; Methodology. **James Legg**: Resources; Investigation; Methodology. **Andrew J Pierce**: Resources; Investigation; Methodology. **Hugo Arasanz**: Resources; Investigation; Methodology. **Gonzalo Fernandez-Hinojal**: Investigation; Methodology. **Ruth Vera**: Resources; Investigation; Methodology. **Karina Ausin**: Resources; Data curation; Software; Validation; Investigation; Methodology. **Enrique Santamaria**: Resources; Data curation; Software; Formal analysis; Supervision; Validation; Investigation; Methodology. **Joaquin Fernandez-Irigoyen**: Resources; Data curation; Software; Formal analysis; Supervision; Validation; Investigation; Methodology. **Grazyna Kochan**: Resources; Formal analysis; Supervision; Funding acquisition; Validation; Investigation; Methodology; Project administration; Writing—review and editing. **David Escors**: Conceptualization; Resources; Data curation; Software; Formal analysis; Supervision; Funding acquisition; Validation; Investigation; Visualization; Methodology; Writing—original draft; Project administration; Writing—review and editing.

Source data underlying figure panels in this paper may have individual authorship assigned. Where available, figure panel/source data authorship is listed in the following database record: biostudies:S-SCDT-10_1038-S44321-024-00098-y.

## Disclosure and competing interests statement

CJE, JL and DE are inventors of the Humabody CB213 (WO/2019/158942. Crescendo Biologics Ltd.). The rest of the authors declare no competing interests.

# Expanded View Figures

**Figure EV1.  Correlation of *PDCD1* and *LAG3* expression with immune cell infiltrates.**

(**A**) Heatmap of partial purity-adjusted Spearman's correlates calculated with TIMER 2.0. between *PDCD1/LAG3* expression and lymphoid infiltrates in a total number of 12159 samples distributed on TCGA cancers. (**B**) Heatmap of partial purity-adjusted Spearman's correlates calculated with TIMER 2.0. between *PDCD1/LAG3* expression and non-lymphoid infiltrates in a total number of 12159 samples distributed on TCGA cancers.

▶

                                           

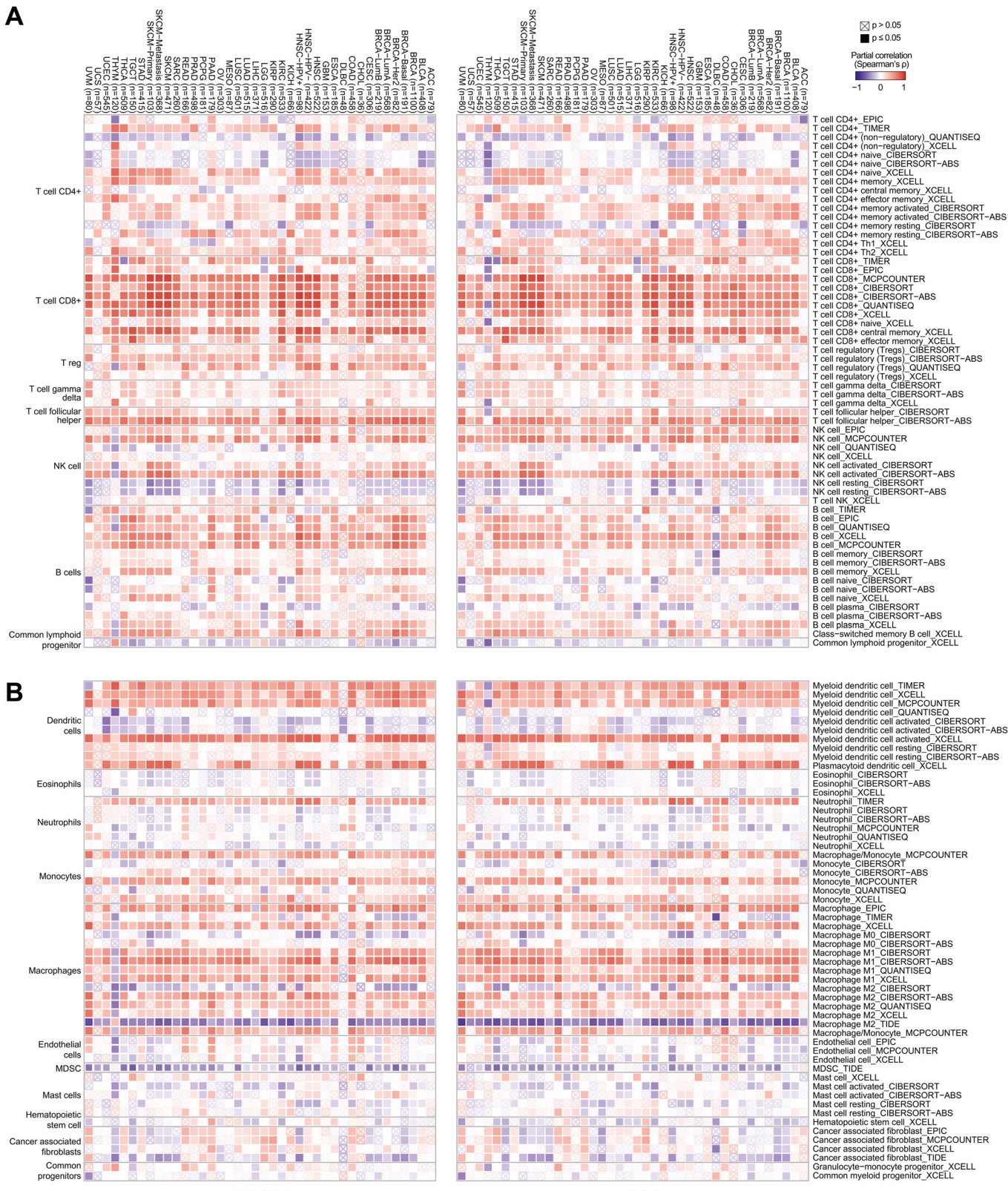

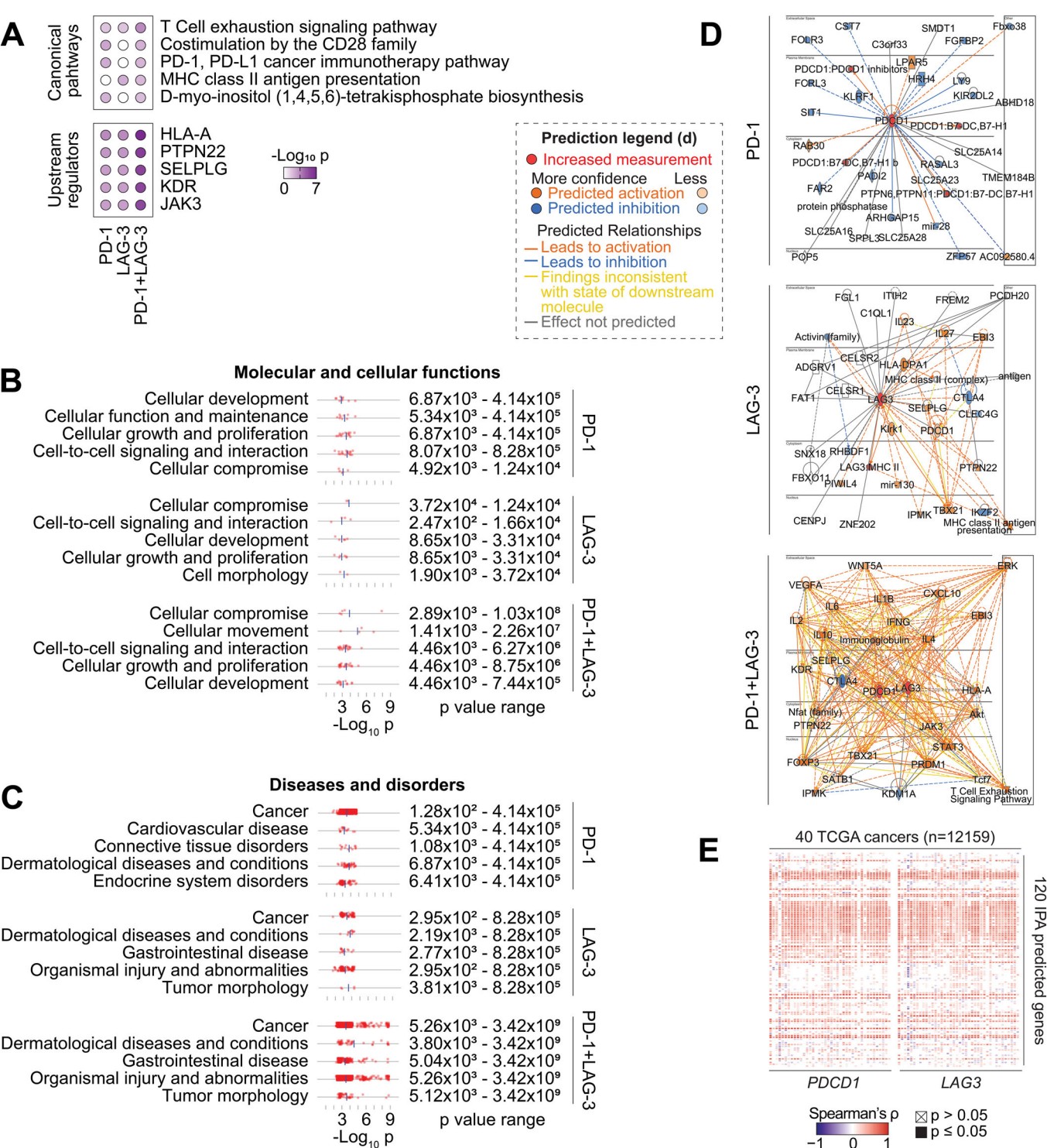

◄ **Figure EV2.  Regulatory networks and causal relationships associated with PD-1/LAG-3 signature.**

(A) Identified enriched canonical pathways and upstream regulators for the upregulation of PD-1/LAG-3 and combinations. (B) Identified enriched molecular and cellular functions for the upregulation of PD-1/LAG-3 and combinations. (C) Identified enriched diseases and disorders for the upregulation of PD-1/LAG-3 and combinations. (D) Predicted regulatory interactomes and associated networks with the indicated PD-1 and LAG-3 signatures. Key nodes are shown, and inter-nodal lines represent functional relationships between nodes. In red, upregulated input molecules as indicated (PD-1 and LAG-3). In blue, downregulated input molecules as indicated (PD-1 and LAG-3). Blue lines, predicted inhibition; orange lines, predicted activation; grey indicates a predicted relationship with a non-predicted effect, and yellow lines, predicted relationship findings inconsistent with the state of the downstream molecule. (E) Heatmap of partial purity-adjusted Spearman's correlates calculated with TIMER 2.0. between *PDCD1/LAG3* expression and a selection of genes regulating identified by IPA of a total number of 12159 samples distributed on the indicated TCGA cancers. Data information: For (A–D), QIAGEN IPA algorithms were used (accessed on 2024), applied on data from curated publicly available datasets of RNA-seq, small RNA-seq, metabolomics, proteomics, microarrays including miRNA and SNP, and small-scale experiments. IPA utilizes two scores for inference; *P*-values from a Fisher's exact test to obtain an enrichment score, and a Z-score to assess the match of observed and predicted regulation patterns.

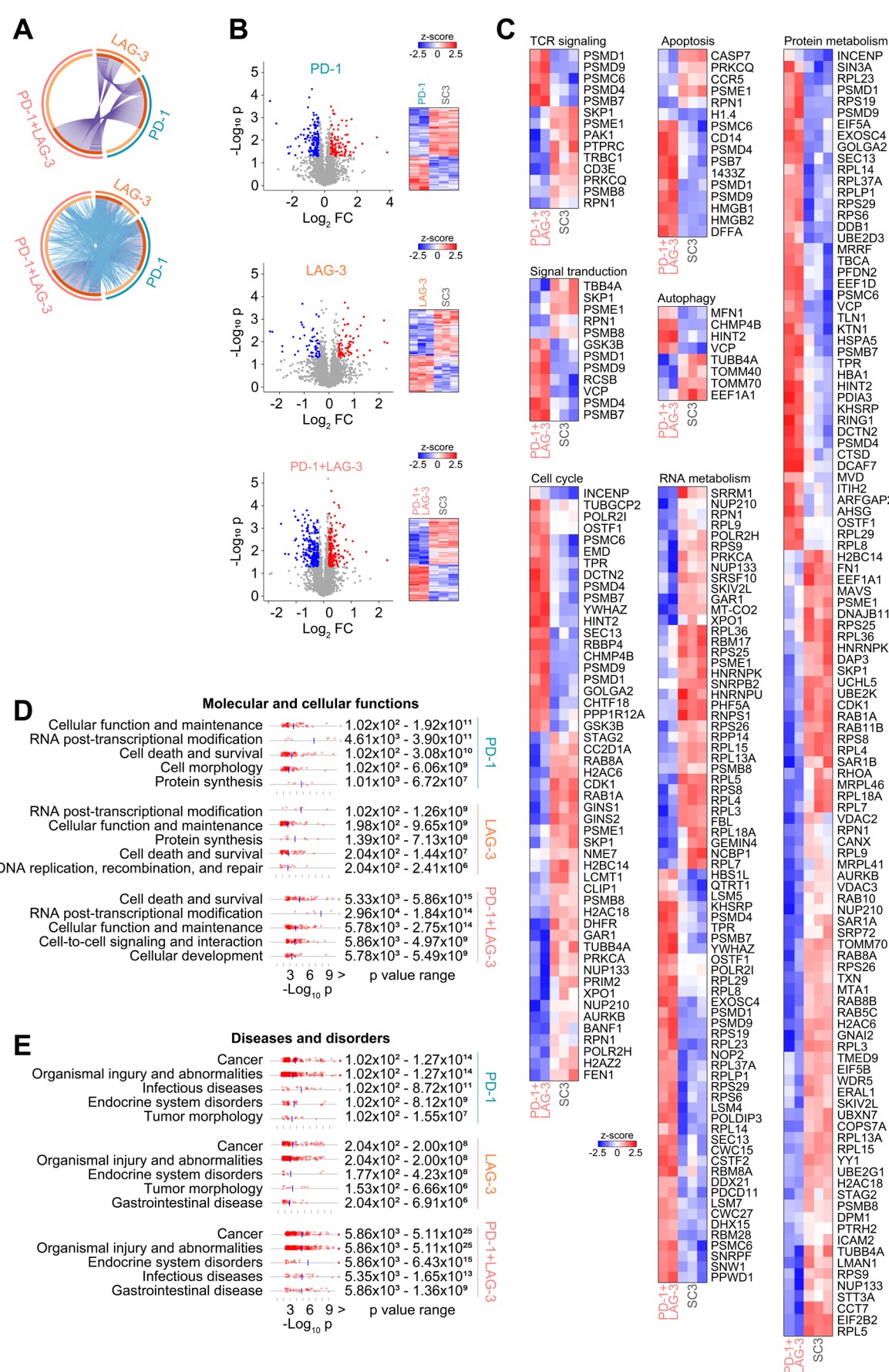

**Figure EV3.  Proteomes of T-cells with PD-1/LAG-3-regulated pathways.**

(**A**) Circo plot representing the overlap from the input proteome dataset lists. Upper circle: On the outside, each arc represents the identity of each proteome. On the inside, each arc represents a list, where each gene has a spot on the arc. Dark orange color represents the proteins that appear in multiple lists and light orange color represents proteins that are unique to that list. Purple lines link the same protein that are shared by multiple lists. Above circle: On the outside, same as upper circle. On the inside, each arc represents a list, where each gene has a spot on the arc. Dark orange color represents the molecules that appear in multiple lists and light orange color represents molecules that are unique to that list. Purple lines link the same gene that are shared by multiple lists. Blue lines link the different genes where they fall into the same ontology term (the term has to statistically significantly enriched and with size no larger than 100). Blue links indicate the degree of functional overlap among the input lists. (**B**) Volcano plots and heatmap with the number of differentially regulated proteins in PD-1, LAG-3, and PD-1 + LAG-3 Jurkat T-cell lines compared to SC3 control cells ($p$-value $\leq$ 0.05) for upregulated (Red, $Log_2$ (Fold Change) $\geq$ 0.38) and downregulated (Blue, $Log_2$ (Fold Change) $\leq$ −0.38) proteins. Blue: significantly downregulated, red: significantly downregulated. Grey: not significantly regulated. (**C**) Heatmaps of differential protein expression in the PD-1 + LAG-3 proteomic dataset compared with the proteome of the SC3-Jurkat control cell line. Red, significantly upregulated proteins ($p$-value $\leq$ 0.05, $Log_2$ (Fold Change) $\geq$ 0.38); blue, significantly downmodulated proteins ($p$-value $\leq$ 0.05, $Log_2$ (Fold Change) $\leq$ −0.38). Relevant T-cell pathways and functions are indicated on top. Specific targets are indicated on the right. (**D**) Identified enriched molecular and cellular functions for the PD-1/LAG-3 proteomes. (**E**) Identified enriched diseases and disorders for the PD-1/LAG-3 proteomes. Data information: Statistical comparisons are shown in the graph as indicated in Methods. For (**B**, **C**), Perseus was used for statistical analyses. An unpaired Student t-test was used for direct comparisons between two groups of samples. Differential PD-1/LAG-3 proteins versus the SC3 control condition comparisons were identified, following $p$-value $\leq$ 0.05, $Log_2$ (Fold Change) $\geq$ 0.38 and $Log_2$ (Fold Change) $\leq$ −0.38 criteria. For (**D**, **E**), QIAGEN IPA algorithms were used (accessed on 2024), applied on data from curated publicly available datasets of RNA-seq, small RNA-seq, metabolomics, proteomics, microarrays including miRNA and SNP, and small-scale experiments. IPA utilizes two scores for inference; $P$-values from a Fisher's exact test to obtain an enrichment score, and a Z-score to assess the match of observed and predicted regulation patterns", as indicated in the rest of the legends. It is lacking the last part of the sentence by mistake, sorry for the inconvenience. Source data are available online for this figure.

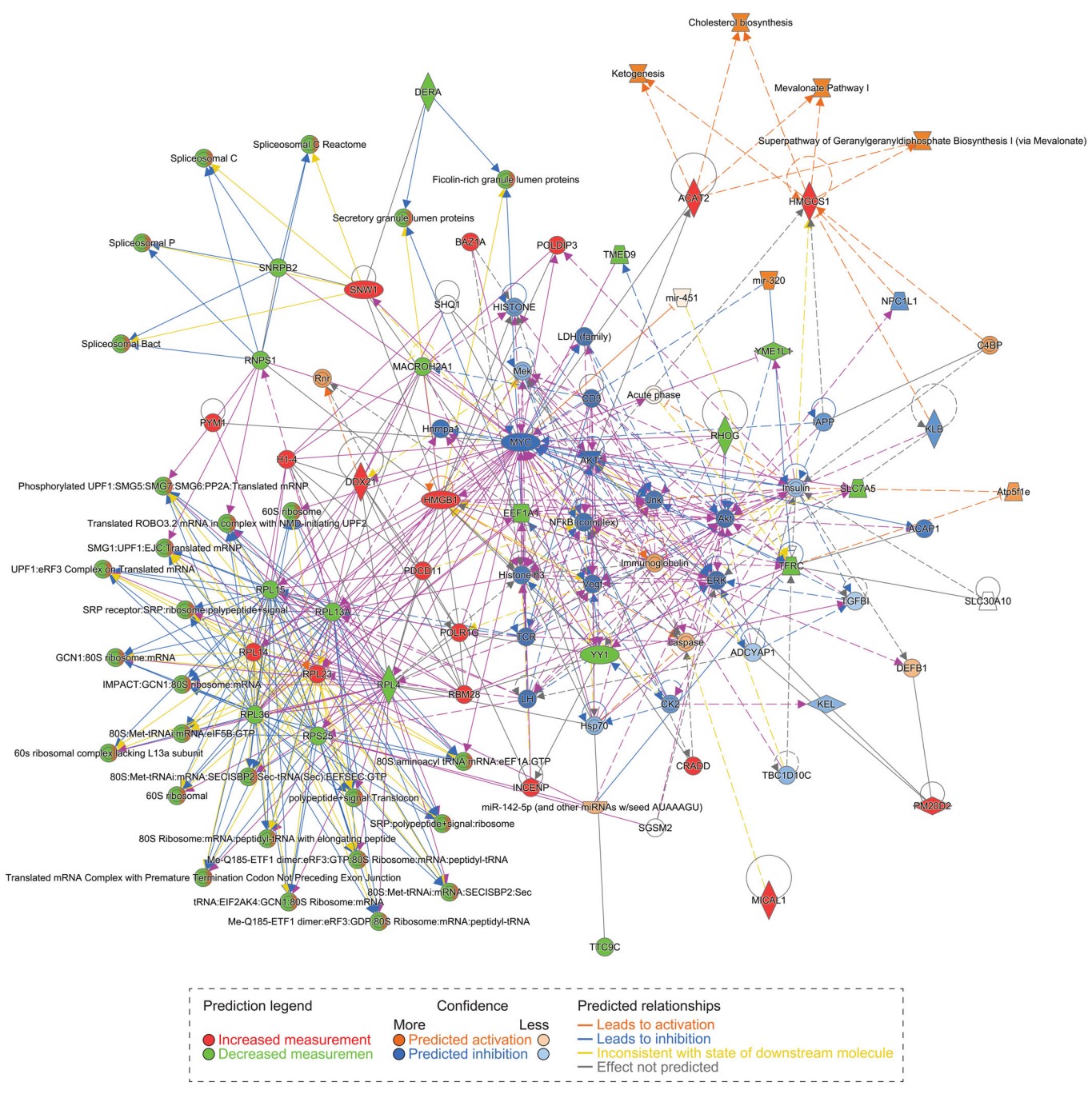

**Figure EV4.  Proteomic interactome networks associated to constitutive activation of PD-1/LAG-3 in Jurkat T-cell lines, generated by IPA.**

Top networks describing potential molecular interactions of the 35 commonly regulated dataset molecules associated to the 35 commonly regulated proteins. In red, upregulated proteins. In green, downregulated proteins. Blue lines, predicted inhibition; orange lines, predicted activation; grey indicates a predicted relationship with a non-predicted effect, and yellow lines, predicted relationship findings inconsistent with the state of the downstream molecule. The specific legends to inter-nodal relationships are described in IPA (Ingenuity Pathway Analysis | QIAGEN Digital Insights). QIAGEN IPA algorithms were used (accessed on 2024), applied on data from curated publicly available datasets of RNA-seq, small RNA-seq, metabolomics, proteomics, microarrays including miRNA and SNP, and small-scale experiments. IPA utilizes two scores for inference; *P*-values from a Fisher's exact test to obtain an enrichment score, and a Z-score to assess the match of observed and predicted regulation patterns.

