## [Peer Review File · EMBO Molecular Medicine]

PD-1/LAG-3 co-signaling profiling uncovers CBL ubiquitin ligases as key immunotherapeutic targets

Luisa Chocarro, Ester Blanco, Leticia Fernandez-Rubio, Maider Garnica, Miren Zuazo, Maria Garcia, Ana Bocanegra, Miriam Echaide, Colette Johnston, Carolyn Edwards, James Legg, Andrew Pierce, Hugo Arasanz, Gonzalo Fernandez-Hinojal, Ruth Vera, Karina Ausin, Enrique Santamaria, Joaquin Fernandez-Irigoyen, Grazyna Kochan, and David Escors

Corresponding authors: David Escors (descorsm@navarra.es) , Luisa Chocarro (luisa.chocarro.deerauso@navarra.es)

Review Timeline:

Submission Date:	12th Jan 24
Editorial Decision:	12th Feb 24
Revision Received:	2nd May 24
Editorial Decision:	4th Jun 24
Revision Received:	13th Jun 24
Accepted:	21st Jun 24

Editor: Lise Roth

Transaction Report:

12th Feb 2024

Dear Dr. Escors,

Thank you for the submission of your manuscript to EMBO Molecular Medicine. We have now received feedback from the three reviewers who agreed to evaluate your manuscript. As you will see from the reports below, the referees acknowledge the interest of the study and are overall supporting publication of your work pending appropriate revisions. Please note that all referees mentioned the need for re-writing the manuscript to improve the presentation of the results and general flow of the manuscript.

Adequately addressing this aspect as well as the other reviewers' concerns in full will be necessary for further considering the manuscript in our journal, and acceptance of the manuscript will entail a second round of review. EMBO Molecular Medicine encourages a single round of revision only and therefore, acceptance or rejection of the manuscript will depend on the completeness of your responses included in the next, final version of the manuscript. For this reason, and to save you from any frustrations in the end, I would strongly advise against returning an incomplete revision.

We are expecting your revised manuscript within three months, if you anticipate any delay, please contact us.

We require:

- 1) A .docx formatted version of the manuscript text (including legends for main figures, EV figures and tables). Please make sure that the changes are highlighted to be clearly visible.
- 2) Individual production quality figure files as .eps, .tif, .jpg (one file per figure). For guidance, download the 'Figure Guide PDF' (<https://www.embopress.org/page/journal/17574684/authorguide#figureformat>).
- 3) At EMBO Press we ask authors to provide source data for the main figures. Our source data coordinator will contact you to discuss which figure panels we would need source data for and will also provide you with helpful tips on how to upload and organize the files.
- 4) A .docx formatted letter INCLUDING the reviewers' reports and your detailed point-by-point responses to their comments. As part of the EMBO Press transparent editorial process, the point-by-point response is part of the Review Process File (RPF), which will be published alongside your paper.
- 5) A complete author checklist, which you can download from our author guidelines (<https://www.embopress.org/page/journal/17574684/authorguide#submissionofrevisions>). Please insert information in the checklist that is also reflected in the manuscript. The completed author checklist will also be part of the RPF.
- 6) Please note that all corresponding authors are required to supply an ORCID ID for their name upon submission of a revised manuscript. An ORCID identifier is currently missing for Luisa Chocarro.
- 7) For data quantification: please specify the name of the statistical test used to generate error bars and P values, the number (n) of independent experiments (specify technical or biological replicates) underlying each data point and the test used to calculate p-values in each figure legend. The figure legends should contain a basic description of n, P and the test applied. Graphs must include a description of the bars and the error bars (s.d., s.e.m.). Please provide exact p values.
- 8) Our journal encourages inclusion of *data citations in the reference list* to directly cite datasets that were re-used and obtained from public databases. Data citations in the article text are distinct from normal bibliographical citations and should directly link to the database records from which the data can be accessed. In the main text, data citations are formatted as follows: "Data ref: Smith et al, 2001" or "Data ref: NCBI Sequence Read Archive PRJNA342805, 2017". In the Reference list, data citations must be labeled with "[DATASET]". A data reference must provide the database name, accession number/identifiers and a resolvable link to the landing page from which the data can be accessed at the end of the reference. Further instructions are available at .
- 9) For more information: There is space at the end of each article to list relevant web links for further consultation by our readers.

Could you identify some relevant ones and provide such information as well? Some examples are patient associations, relevant databases, OMIM/proteins/genes links, author's websites, etc...

10) Author contributions: CRediT has replaced the traditional author contributions section because it offers a systematic machine readable author contributions format that allows for more effective research assessment. Please remove the Authors Contributions from the manuscript and use the free text boxes beneath each contributing author's name in our system to add specific details on the author's contribution. More information is available in our guide to authors.

11) Disclosure statement and competing interests: We updated our journal's competing interests policy in January 2022 and request authors to consider both actual and perceived competing interests. Please review the policy <https://www.embopress.org/competing-interests> and update your competing interests if necessary.

12) Every published paper now includes a 'Synopsis' to further enhance discoverability. Synopses are displayed on the journal webpage and are freely accessible to all readers. They include a short stand first (maximum of 300 characters, including space) as well as 2-5 one-sentences bullet points that summarizes the paper. Please write the bullet points to summarize the key NEW findings. They should be designed to be complementary to the abstract - i.e. not repeat the same text. We encourage inclusion of key acronyms and quantitative information (maximum of 30 words / bullet point). Please use the passive voice. Please attach these in a separate file or send them by email, we will incorporate them accordingly.

13) As part of the EMBO Publications transparent editorial process initiative (see our Editorial at <http://embomolmed.embopress.org/content/2/9/329>), EMBO Molecular Medicine will publish online a Review Process File (RPF) to accompany accepted manuscripts.

In the event of acceptance, this file will be published in conjunction with your paper and will include the anonymous referee reports, your point-by-point response and all pertinent correspondence relating to the manuscript. Let us know whether you agree with the publication of the RPF and as here, if you want to remove or not any figures from it prior to publication. Please note that the Authors checklist will be published at the end of the RPF.

I look forward to receiving your revised manuscript.

Yours sincerely,

Lise Roth

***** Reviewer's comments *****

Referee #1 (Comments on Novelty/Model System for Author):

the model is not, strictly speaking, inadequate, but its choice is questionable

Referee #1 (Remarks for Author):

The authors have analyzed, through a very detailed and up-to-date study, how PD1 and LAG3 cooperate to generate a

dysfunctional anti-tumor immune response in human cancers.

The question the authors addressed in this paper and the work carried out to answer it are very interesting and contribute to a better understanding of the mechanism of resistance to anti-PD1 monotherapy.

However, there's so much information that it's hard to pick out the most important ones and keep track of it all. This extensive study really needs to be made more digestible for the reader by highlighting the data that best support the core results.

Some figures, in particular Fig 1, 3, 4, are unattractive : too many information, sometimes redundant, not easy to read. I would suggest that the authors reconsider whether each panel of each figure is essential to understanding and how the figure can best convey the desired information. Simplifying them will greatly help to get the information at first glance.

In the result section, notably for the omics analysis, the authors strongly refer to supplementary data, which raises the question of the relevance of including them in supplementary data. This really underlines the need to select the data the authors feel is best suited to illustrate their main findings.

It is now well known that ubiquitination plays a crucial role in regulating the degradation of immune checkpoints and the activation of immune-related pathways, notably PD-1/PD-L1 signaling pathway. So what is the link between all the omics studies described and the targeting of ubiquitination? Doesn't this dilute the take-home messages?

Some points to be clarified

-The tree structures in the heat-maps are not defined.

-What is the information to be retained from figure 4d ?

-If correctly understood, peripheral T cells of NSCLC patients resistant to PD1/PD-L1 blockade therapy highly co-express PD1 and LAG-3 (p10). So why, in in fig 5c, it is necessary to stimulate them ex vivo with A549-SC3? not clear to me.

-What about the expression of CBLB in PD-1/LAG-3 low T cells ?

p12, Fig1g should be replaced by Fig 5g.

-In vivo experiment design : treatments start when tumors are very small. Can the authors explain the choice of this experimental design? Peritoneal injection of antibodies ?

In conclusion, for specialists in the field, the work is of interest, even if, in its current form, it is still too dense. However, for non-specialists, its complexity and density make it difficult to read.

Referee #2 (Comments on Novelty/Model System for Author):

Much of the manuscript uses T cell lines to make their major points with dual expression of LAG3 and PD1 with signaling moieties transfected into them. This gets them to the point where they reveal CBL-B and cCBL as well as other E3 ligases as the important changes.

Referee #2 (Remarks for Author):

This was not an easy manuscript to review given the abundance of cell line and transfection data. Overall, the conclusions are sound and support those of many other papers, regarding the critical role of CBL-B in limiting the antitumor effectiveness of PD1+LAG3+ cells. Two major comments:

1) In the introduction, the convergent role of CD28 signaling in PDA and CD226/TIGI/CBL-B should be referenced.

[https://www.cell.com/immunity/pdf/S1074-7613\(20\)30404-0.pdf](https://www.cell.com/immunity/pdf/S1074-7613(20)30404-0.pdf)

2) Deeper findings related to LAG3 from the Vignali Group should also be referenced and discussed. Aggarwal V, Workman CJ, Vignali DAA. LAG-3 as the third checkpoint inhibitor. Nat Immunol. 2023 Sep;24(9):1415-1422. doi: 10.1038/s41590-023-01569-z. Epub 2023 Jul 24. PMID: 37488429.

Referee #3 (Comments on Novelty/Model System for Author):

Please see my comments to the authors below.

Referee #3 (Remarks for Author):

Summary

In this study, the authors utilized genomic, proteomic, bioinformatic, in vitro and in vivo tools in combination with publicly and in-house generated datasets to characterize the consequences of PD1 and LAG3 inhibitory receptor signaling in T cells. This is an important and topical question in the field of immunotherapy, as the mechanisms underlying the efficacy of PD1/LAG3 blockade are incompletely understood. Further, not all patients respond to dual PD1/LAG3 blockade and identifying new therapeutics in this space is important. From their analyses, the authors conclude that PD1 and LAG3 influence many aspects of T cell function and highlight negative regulation of TCR signaling and upregulation of E3 ubiquitin ligases. Focusing on the E3 ubiquitin ligase CBL-B, the authors show in an immunotherapy refractory murine model that pharmacological inhibition of CBL-B in combination with dual blockade of PD1 and LAG3 leads to improved survival. The authors conclusions are generally well supported by their data, but a careful reading of the manuscript reveals several important issues that should be addressed as outlined below.

Major Comments

1. In the introduction, the authors discuss the influence of PD1 on T cell receptor signaling. However, there is conflicting evidence as to whether PD1 has a dominant effect on TCR signaling or co-stimulatory signaling (Hui et al Science 2017; Kamphorst et al Science 2017). The authors obviously uncover a TCR centric phenotype in the in vitro system they use, but is there any evidence for modulation of pathways involved with co-stimulatory signaling? If not, this should be mentioned in the discussion as a shortcoming of the in vitro assays and proteomics.
2. In the in vitro system described in Figure 2, the authors find that ZAP70 is downregulated in the presence PD1, LAG3 and PD1 + LAG3. This is likely a consequence of long-term cell culture, and the findings ex vivo in CD8+ T cells are more likely to be reflected in differences in phosphorylation in the TCR signaling cascade such as pZAP70, pSLP76, pERK and pAKT. The authors should evaluate whether blockade of PD1 or LAG3 alone or in combination influences the ability to signaling through the TCR rather than the amount of the signaling proteins that are present. This is a more physically relevant experiment compared to what is presented currently in Figure 5.
3. The authors are to be commended for analyses of specimens from human patients in their manuscript. However, the details associated with some of these experiments are sparse. For example, in Figure 5d-f, are these gated on specific populations or total CD4+ or CD8+ T cells? How long were these cultures performed? Are naïve and memory T cells included or just effector cells? If all T cell subsets are included, it is surprising that CBL-B MFI would be so drastically changed.
4. For all CD4+ T cell analyses, it would be ideal if the authors were to separate conventional CD4+ T cells and regulatory CD4+ T cells. CBL-B may be differentially modulated in these two subpopulations.
5. The authors focus on YY1 in Figure 3, but DDX21 is also highlighted. Why was YY1 selected? Is DDX21 or any of the other proteins indicated by analysis in Figure 5g associated with response to immunotherapy as in Figure 5j? Also, why did the author look at aggregated T cells rather than at least CD4+ and CD8+ T cells?
6. There is a logical jump in Figure 4 that this reviewer finds difficult to follow. The authors identify canonical pathways, upstream regulators and causal networks associated with PD1, LAG3 and PD1+LAG3. Then in Figure 4e the authors present an analysis of E3 ubiquitin ligases with minimal motivation as to why they are assessing them. They then proceed to investigate CBLs, but CBLs aren't included anywhere in the figure. The authors should better explain the logic of this figure that results in a desire to analyze CBL-B.
7. The authors are to be commended for performing analyses in a murine model to evaluate the hypothesis that CBL-B inhibition would improve antitumor immunity. Despite a significant improvement in overall survival with the addition of a CBL-B inhibitor to anti-PD1+anti-LAG3, it would be ideal to understand the cell type(s) responsible for this efficacy. It could potentially be improvement in CD8+ T cell or conventional CD4+ T cell effector function or conversely a reduction in regulatory CD4+ T cell function or both. The authors should address this important mechanistic point.

Minor Comments

1. Line numbers would be helpful for calling out specific corrections.
2. Page 4, first paragraph: "Hence, PD-1 and LAG-3 cooperatively establish a strong dysfunctional estate", estate should be "state".
3. Page 4, first paragraph: "PD-1 and LAG-3 stablish a complex together", stablish should be "establish".
4. Page 14, first paragraph: "This evidenced the association of the PD1/LAG3signature with potentially", evidenced should be "demonstrated" or something similar.
5. Page 14, second paragraph: "The engineering of PD-1+LAG-3 T-cell lines uncovered several T-cell dysfunctions associated to this signature." "To" this signature should be "with" this signature.
6. The last paragraph of the discussion is unnecessary and could be mentioned elsewhere in the discussion. It would be stronger to end on the preceding paragraph.
7. Methods, under flow cytometry: several of the Greek letters do not appear and are instead replaced by empty boxes.
8. Mention of "tox" lists is confusing, given the role of the gene TOX in CD8+ T cell exhaustion.
9. The authors erroneously refer to Figure 5g as Figure 1g in the results.

Recommendation

Major revisions based on additional experiments required to address the points outlined above.

Point-by-Point responses.**REFEREE 1**

1. The authors have analyzed, through a very detailed and up-to-date study, how PD1 and LAG3 cooperate to generate a dysfunctional anti-tumor immune response in human cancers. The question the authors addressed in this paper and the work carried out to answer it are very interesting and contribute to a better understanding of the mechanism of resistance to anti-PD1 monotherapy. However, there's so much information that it's hard to pick out the most important ones and keep track of it all. This extensive study really needs to be made more digestible for the reader by highlighting the data that best support the core results. Some figures, in particular Fig 1, 3, 4, are unattractive : too many information, sometimes redundant, not easy to read. I would suggest that the authors reconsider whether each panel of each figure is essential to understanding and how the figure can best convey the desired information. Simplifying them will greatly help to get the information at first glance.

We sincerely appreciate the Reviewer's comments. Yes indeed, the information is complex and large, but rather of major importance to the field. It was truly an effort to reduce all the information to the figures shown in the paper, although as Reviewer 1 rightly says, it is difficult to read. Hence, we have taken on board the Reviewer's suggestion and we have gone through every figure one by one, and we have undertaken the following steps to simplify them:

- Many of the bar graphs used in the figures (see final figures 1, 3, 4 and EV2) have been replaced by "density dots" type of graphs. These graphs have recently substituted the more classical representations because (i) they significantly reduce space, and (ii) they are by far easier to interpret and understand. In addition, exact p-values and z-scores for each density plot have been added as source data linked to the figure.
- We have removed from each figure the panels less important or redundant panels as suggested. Former Figure 3e is redundant with former Figure 3f. The causal networks can be redundant with upstream regulators, and hence, the former Fig 3i networks have been summarized and merged as supplementary figure EV3; Former figures Fig 4a-c, Fig EV2 a-f, and Fig EV3 d-e have been summarized and simplified as they are redundant. Fig EV4 has been removed as it is redundant with former figure EV3c, and difficult to interpret.

2. In the result section, notably for the omics analysis, the authors strongly refer to supplementary data, which raises the question of the relevance of including them in supplementary data. This really underlines the need to select the data the authors feel is best suited to illustrate their main findings.

This issue is related to the previous one, and derives from the complexity of the data. We agree with the Reviewer, and in our attempt of “no leaving anything out” we placed in supplementary data the gene/protein lists with the numerical data in case that a particular researcher wants to check it. The main figures just reflect the conclusions that we want to draw, but we were referring to supplementary data to provide the reader the opportunity of checking individual genes/proteins within our data. But again, we have provided these data in a data repository and appendix tables and refer to it within the text. Nevertheless, we have undertaken the following steps to simplify the presentation:

- We have removed the following supplementary figures because they are not sufficiently relevant. Former Fig EV2 a-f and Fig EV3 d-e have been summarized and simplified as they were less important or redundant. Former Fig EV4 has been removed as it was redundant with former figure EV3c, and difficult to read.
- We have deposited the proteomic data to the following data repository: Reference: <https://www.ncbi.nlm.nih.gov/pmc/articles/PMC5210561/>) with the identifier PXD040408 for ProteomeXchange and JPST002062 for jPOST, and the immune infiltrate TCGA data as appendix tables. Therefore, we it is stated in the text the following sentences referring to the specific gene/protein datasets:
 - DATA AVAILABILITY: Mass-spectrometry data and search results files were deposited in the Proteome Xchange Consortium via the JPOST partner repository (<https://repository.jpostdb.org/>; Reference: <https://www.ncbi.nlm.nih.gov/pmc/articles/PMC5210561/>) with the identifier PXD040408 for ProteomeXchange and JPST002062 for jPOST (for reviewers: <https://repository.jpostdb.org/preview/118930299263fc6310a6249> ; Access key: 7060).
 - Appendix Table S1 and Appendix Table S2 are cited in the manuscript.

3. It is now well known that ubiquitination plays a crucial role in regulating the degradation of immune checkpoints and the activation of immune-related pathways, notably PD-1/PD-L1 signaling pathway. So what is the link between all the omics studies described and the targeting of ubiquitination? Doesn't this dilute the take-home messages?

We thank the Reviewer for his comment. We agree with the Reviewer that due to the complexity of our manuscript, sometimes it is difficult to follow the rationale behind it. That is why the first paragraph of results highlighted the three main points of our study. We used the omics studies to identify the regulatory pathways that were activated/deactivated in cancer-associated T cells. Then, with this information we reconstructed key regulatory pathways in these T cells, and we underwent a process of identifying from these pathways suitable candidates that could be pharmacologically targeted. One of such is the ubiquitin ligase pathways, but it could have been any other. Due to the Reviewer's comment, we feel that this concept needs to be reinforced in our manuscript. Therefore, we have added the following sentence as a first paragraph of the discussion:

- “Here we carried out thorough and systematic omic studies at multiple levels to identify the regulatory pathways that were activated/deactivated in PD-1/LAG-3+ cancer-associated T cells. This information was used to construct the key regulatory pathways, identifying those that were suitable candidates for pharmacological intervention. The E3 ubiquitin pathway was selected, although our study uncovered several other pathways that could be pharmacologically intervened.”

4. The tree structures in the heat-maps are not defined.

We have defined in the manuscript the tree structures in the Fig 3 and 4 legends as follows: Heatmap tree structures represent hierarchical clustering based on Euclidean distances.

5. What is the information to be retained from figure 4d?

We apologise to the Reviewer for not being sufficiently clear. This graph represents the convergence of distinct molecules in our proteomic database that are regulated by the inhibition of the TCR function in an upstream manner. Upstream regulator analysis was performed on Ingenuity Pathway analysis, that uses a statistical approach to determine TCR as an upstream regulator and to score the network connections and regulation directions towards our dataset differential proteins. We agree that considering the whole manuscript this figure might seem redundant. To clarify this point we have explained the overall conclusion of this graph in the text manuscript in page 11: “Fig 4b summarises the finding of TCR as an inhibited upstream regulator (in blue) of the differential PD-1/LAG-3 proteomic dataset, explaining the convergence of the observed expression changes (in red and green).”.

6. If correctly understood, peripheral T cells of NSCLC patients resistant to PD1/PD-L1 blockade therapy highly co-express PD1 and LAG-3 (p10). So why, in in fig 5c, it is necessary to stimulate them ex vivo with A549-SC3? not clear to me. What about the expression of CBLB in PD-1/LAG-3 low T cells ? Page 12, Fig1g should be replaced by Fig 5g.

We sincerely apologise to the Reviewer. Due to the complexity of our manuscript, we referenced our previous publications without explaining in detail our experimental systems. Peripheral blood T cells from NSCLC cancer patients resistant to PD1/PDL-1 blockade highly co-express PD1 and LAG-3 following TCR stimulation. Therefore, in Zuazo et al. EMBO Mol Med. 2019 we set up a system that mimicked the interaction between cancer cells and T-cells through TCR stimulation. This TCR stimulation is needed need to (i) Co-upregulate PD-1 and LAG-3 (described in our work by Zuazo et al. EMBO Mol Med. 2019), and (ii) induce transcriptional transactivation of E3 ubiquitin ligases, that takes about 1 day to induce TCR-downmodulation (described in our paper by Karwacz et al. EMBO Mol Med. 2011). PD-1low (with or without LAG-3low) T cells do not express CBL-B.

To clarify the points raised by the Reviewer, we added the following in the manuscript text:

- Page 12: “We previously demonstrated in an ex vivo A549 cancer cell-T cell interaction assay that peripheral blood T cells from NSCLC cancer patients need to be TCR stimulated to co-upregulate PD-1 and LAG-3 (Zuazo et al., 2019), and to induce the transcriptional transactivation of E3 ubiquitin ligases (Karwacz et al., 2011). Briefly, human A459 lung adenocarcinoma cells were engineered to express a membrane bound anti-CD3 single-chain antibody (SC3) to stimulate T cells from NSCLC patients through a 4-day co-culture.”
- Fig 1g reference refers to the modelling of PD-1/LAG-3 co-expression signature that was previously presented on the previous results sections. We apologize to the reviewer for the redundance, so reference to the figure in this later part of the manuscript has been removed.

7. In vivo experiment design : treatments start when tumors are very small. Can the authors explain the choice of this experimental design? Peritoneal injection of antibodies.

We appreciate the points raised by the Reviewer, so we can clarify them. This experimental set up is standard in immunotherapy research with transplantable tumors. Immune checkpoint blockade is based on stimulating T-cells by systemic administration of ICI antibodies. However, unlike targeted therapies or chemotherapy, it takes about 10 to 15 days to raise a T-cell response against the tumor, which

depends on clonal expansion, infiltration and cytotoxicity at the tumor site. This was demonstrated by our group some years ago (Karwacz et al. 2011. EMBO Mol Med; Karwacz et al. 2012. Oncoimmunology). Therefore, tumors are allowed to grow to a certain point up to when there is still time to raise an effective T cell response. Otherwise, the critical size is reached and animals have to be humanly sacrificed. The specific maximum tumor size to detect immunotherapeutic effects also depends on the growth kinetics of the specific model.

Peritoneal injection of antibodies is also standard in mouse models, and in immune checkpoint blockade strategies in rodents. In humans, antibodies are systemically administered through intravenous injection about once every two to four weeks. In mice, due to metabolic and pharmacokinetic differences, the injections have to take place at least twice every week. This would imply intravenous injections through the tail vein every two or three days, which is unacceptable from an ethical point of view. Therefore, it is substituted by intraperitoneal injection of ICI antibodies, which also achieves systemic distribution. Indeed, this administration route is also used to systemically deplete immune cell populations.

Nevertheless, due to the Reviewer's comments, we realized that these points need to be specified for the non-specialized reader. Hence, we added the following in the manuscript text:

- In page 19: "Anti-tumor T-cell responses take a minimum of 10 days to act (Karwacz et al., 2011). Therefore, tumors were allowed to grow up to an average diameter of 3.5 mm (day 0), before starting immunotherapies, as the growth of Lacun3 tumors show fast kinetics."
- In page 19: "Antibodies were administered intraperitoneally following standard procedures in mouse immunotherapy models"

REFEREE 2

Referee 2 also states that conclusions are sound, but the manuscript was not easy to read. The Referee made 2 points, that we have fully addressed as follows:

Much of the manuscript uses T cell lines to make their major points with dual expression of LAG3 and PD1 with signaling moieties transfected into them. This gets them to the point where they reveal CBL-B and cCBL as well as other E3 ligases as the important changes.

1. In the introduction, the convergent role of CD28 signaling in PDA and CD226/TIGI/CBL-B should be referenced. [https://www.cell.com/immunity/pdf/S1074-7613\(20\)30404-0.pdf](https://www.cell.com/immunity/pdf/S1074-7613(20)30404-0.pdf)

We agree with the Reviewer, but we believe that the role of CBL-B in PD-1, LAG-3, CD28 and TIGIT would be more suited for the discussion section. We have included the potential major role of CBL proteins in other resistance mechanisms as follows:

- Page 16: “Notably, ubiquitination via CBL-B and proteasomal degradation may play a major role not only on PD-1/LAG-3 signaling but also on the post-transcriptional regulation of other immune checkpoints such as CD226, TIGIT or CD28 related signaling molecules (Braun et al, 2020).”

2. Deeper findings related to LAG3 from the Vignali Group should also be referenced and discussed. Aggarwal V, Workman CJ, Vignali DAA. LAG-3 as the third checkpoint inhibitor. Nat Immunol. 2023 Sep;24(9):1415-1422. doi: 10.1038/s41590-023-01569-z. Epub 2023 Jul 24. PMID: 37488429.

We agree with the Referee and we have expanded the discussion of LAG-3 functions including the suggested reference for the reader as follows:

- In page 14,: (Burova et al, 2019; Chocarro et al, 2022b; Ghosh et al, 2019; Jiang et al, 2021; Sordo-Bahamonde et al, 2021; Tawbi et al, 2022; Zahm et al, 2021). Importantly, although LAG-3 has been extensively described as a key potential immunomodulatory molecule, its mechanism of action remains to be controversial and further studies are required to characterize its signaling in T cells (Aggarwal et al, 2023)..

REFEREE 3

Referee three thinks that our manuscript is important for the immunotherapy field. T cells, and points out some issues that need addressing. We have addressed all the issues as follows:

1. In the introduction, the authors discuss the influence of PD1 on T cell receptor signaling. However, there is conflicting evidence as to whether PD1 has a dominant effect on TCR signaling or co-stimulatory signaling (Hui et al Science 2017; Kamphorst et al Science 2017). The authors obviously uncover a TCR centric phenotype in the in vitro system they use, but is there any evidence for modulation of pathways involved with co-stimulatory signaling? If not, this

should be mentioned in the discussion as a shortcoming of the in vitro assays and proteomics.

We appreciate the comment from the Reviewer. PD-1 plays key roles in signalling at different levels. First, during antigen presentation to naïve T cells, that require in most instances CD28-CD80 stimulation. Then, at the tumor site, where in most instances CD28-CD80 interactions are not going to take place between cancer cells and T cells, as effector T cells only require TCR stimulation through peptide-MHC binding to exert cytotoxicity. In this context PD1-PDL1 interactions will interfere with TCR signaling in the absence of CD28 interactions. The Reviewer's suggestion is a fair point. We need to specify in the manuscript that the signaling studied in our study is CD28-independent. To clarify this point, we added in the discussion the following paragraph:

- Page 15,: "PD-1 plays key roles in signaling at different levels (Hui et al Science 2017; Kamphorst et al Science 2017; Karwacz et al. 2011. EMBO Mol Med). First, during antigen presentation by professional antigen presenting cells to T cells, that requires in most instances CD28-CD80 costimulation. Then, at the tumor site, where CD28-CD80 interactions are in most instances absent between cancer cells and T cells, as effector T cells only require TCR stimulation through peptide-MHC binding to exert cytotoxicity. In this context PD1-PDL1 interactions would interfere with TCR signaling in the absence of CD28 interactions. It needs to be remarked that the assays utilized here do not include CD28 signaling. Our data may better reflect the mechanisms taking place within the tumor environment but not during antigen presentation by professional antigen presenting cells, where PD-1-PD-L1 signaling plays a major regulatory role as shown before (Karwacz et al., 2011)..

Nevertheless, we analysed the data associated to CD28 signaling, because it could still be possible that the mechanism potentially interfering with CD28 signal transduction may still be there. Therefore, we found that CD28 was identified by Ingenuity Pathway Analysis as an inhibited upstream regulator for all PD-1, LAG-3 and PD-1+LAG-3 differential proteomes

In this paper, we used the omics studies to identify the regulatory pathways that were activated/deactivated in cancer-associated T cells. Then, with this information we reconstructed key regulatory pathways in these T cells, and we underwent a process of identifying from these pathways suitable candidates that could be pharmacologically targeted. One of such is the ubiquitin ligase pathways, but it could have been any other. Due to the Reviewer's 1 and 3 comments, we feel that this concept needs to be reinforced in our manuscript. Therefore, we have added the following sentence as a first paragraph of the discussion:

- Page 14: “Here we carried out thorough and systematic omic studies at multiple levels to identify the regulatory pathways that were activated/deactivated in PD-1/LAG-3+ cancer-associated T cells. This information was used to construct the key regulatory pathways, identifying those that were suitable candidates for pharmacological intervention. The E3 ubiquitin pathway was selected, although our study uncovered several other pathways that could be pharmacologically intervened.”

2. In the in vitro system described in Figure 2, the authors find that ZAP70 is downregulated in the presence PD1, LAG3 and PD1 + LAG3. This is likely a consequence of long-term cell culture, and the findings ex vivo in CD8+ T cells are more likely to be reflected in differences in phosphorylation in the TCR signaling cascade such as pZAP70, pSLP76, pERK and pAKT. The authors should evaluate whether blockade of PD1 or LAG3 alone or in combination influences the ability to signaling through the TCR rather than the amount of the signaling proteins that are present. This is a more physically relevant experiment compared to what is presented currently in Figure 5.

The referee suggests that the findings in figure 2 are a consequence of long term cell culture. We disagree with the Reviewer, as these experiments do have their appropriate controls, including unmodified T cells, T cell lines expressing the SC3 control, T cell lines expressing PD-1, T cell lines expressing LAG-3 and T cell lines co-expressing both. The results are clear, as these cultures have been treated simultaneously in the same way throughout the experiments. PD-1 and LAG-3 constitutive signaling induce the decrease in the shown signaling molecules. The data is relevant because it is also present in the proteomics. There is a significant decrease not only in the signaling molecules, but also in the shown markers and cytokines. We need to remark that the data ex vivo shown in Figure 5 is not in T cell lines but in primary T cells, which go along the data from the T cell lines but also from the omics data.

Indeed, we have assessed the abilities of PD-1, LAG-3 and PD-1/LAG-3 co-blockade in the context TCR stimulation in primary T cells, but we are afraid according to the comments of the other referees that this has not been sufficiently explained in our manuscript. The data indeed is in figures 5 using primary T cells from NSCLC patients. This experimental system fully relies on TCR stimulation by co-culture with A549 human lung adenocarcinoma cells expressing a membrane bound anti-CD3 single-chain antibody (SC3). Therefore, we are indeed triggering TCR signaling, which leads to significantly different results if PD-1, LAG-3 or PD-1/LAG-3 blockers are present. And the results are clear, consistent between all the datasets from cancer biopsy data,

between all the T cell lines and primary T cells, and fundamental to the conclusions of the paper. H

However, and to clarify the point raised by the reviewer, we need to stress that in the context of TCR stimulation. To clarify the points raised by the Reviewer and Reviewer 1, we added the following in the manuscript text:

- Page 12,: “We previously demonstrated in an ex vivo A549 cancer cell-T cell interaction assay that peripheral blood T cells from NSCLC cancer patients need to be TCR stimulated to co-upregulate PD-1 and LAG-3 (Zuazo et al., 2019), and to induce the transcriptional transactivation of E3 ubiquitin ligases (Karwacz et al., 2011). Briefly, human A459 lung adenocarcinoma cells were engineered to express a membrane bound anti-CD3 single-chain antibody (SC3) to stimulate T cells from NSCLC patients through a 4-day co-culture.”

We also agree with the reviewer that PD-1/LAG-3 signaling might lead to differences in phosphorylation in the TCR signaling cascade such as pZAP70, pSLP76, pERK and pAKT.

To answer these issues, we would have to release confidential information at this time in the form of a second paper that we are preparing (with intention to submit to EMBO Mol Med as a follow-up). As EMBO Mol Med does not allow to cite “data not shown”, we have not specified these issues more profoundly in the current manuscript.

We can nevertheless release that we have extensive proteomics and phosphoproteomics data, similar to this study but including about 30 different LAG-3 mutants on the signaling domains. **These data elucidates the differences in phosphorylation in the TCR signalling driven by PD-1/LAG-3 signaling.** We would still like to maintain this data confidential for the time being. **However, if the reviewer deems this to be necessary, we are open to release the data to the Prof. Roth, the editor of the current submission, so certify that we do indeed possess all these data.** But for the time being, these data is more fit for a second follow-up paper.

3. The authors are to be commended for analyses of specimens from human patients in their manuscript. However, the details associated with some of these experiments are sparse. For example, in Figure 5d-f, are these gated on specific populations or total CD4+ or CD8+ T cells? How long were these cultures performed? Are naïve and memory T cells included or just effector cells? If all T cell subsets are included, it is surprising that CBL-B MFI would be so drastically changed.

We apologise to the Reviewer, as many of the issues were caused by the complexity of the paper, that led to minimizing some of the experimental details. The results from Figure 5d-f were carried out in total CD4 and total CD8. As we specified in the previous comment, these cultures were not only of T cells, but these were TCR-stimulated using A549 lung cancer cells expressing SC3. Therefore, the co-cultures with or without antibodies were carried out for 4 days. The majority of the T cells in these patient cohorts correspond to effector memory and effector T cells, as we have previously shown in (Zuazo et al. 2019. EMBO Mol Med). It is not surprising that CBL-B MFI is so drastically changed because CBL-B is transcriptionally transactivated by TCR and PD-1 signaling. Our group has been working for a number of years on PD-1 signaling and ubiquitin ligases, and we have obtained comparable results also in mouse T cells. For example, interference with PD-1 signaling drastically reduces CBLB upregulation as we showed previously in mouse T cells in (Karwacz et al. EMBO Mol Med). Indeed, one of the Referees have suggested to clarify this issue.

Therefore, we agree with the Reviewer that these issues need more detail, and we have included in the text the following:

- In Figure 5 legend:...."data from total CD4 and CD8 gated populations..."
- Page 12: "We previously demonstrated in an ex vivo A549 cancer cell-T cell interaction assay that peripheral blood T cells from NSCLC cancer patients need to be TCR stimulated to co-upregulate PD-1 and LAG-3 (Zuazo et al., 2019), and to induce the transcriptional transactivation of E3 ubiquitin ligases (Karwacz et al., 2011). Briefly, human A459 lung adenocarcinoma cells were engineered to express a membrane bound anti-CD3 single-chain antibody (SC3) to stimulate T cells from NSCLC patients through a 4-day co-culture".
- In page 11: "The majority of T cells from the NSCLC patient cohort were effector memory and effector T cells as described in (Zuazo et al., 2019)."
- In results, page 12: PD-1 blockade inhibited CBL-B up-regulation in T-cells from NSCLC patients, as we previously showed in mouse T cells"

4. For all CD4+ T cell analyses, it would be ideal if the authors were to separate conventional CD4+ T cells and regulatory CD4+ T cells. CBL-B may be differentially modulated in these two subpopulations.

We agree with the Reviewer that it would be ideal to have data on Tregs. We indeed looked at them in our cohort of patients in Zuazo et al EMBO Mol Med. 2019. In these patients, the CD4 T cells present highly-differentiated phenotypes, mostly effector memory and effector T cells, with a degree of proliferative anergy. However, Tregs were virtually absent in systemic peripheral blood in our cohort of patients, so we could not perform any experiments on them. Of course, Tregs are enriched within the tumor

microenvironment, but that is challenging to obtain in NSCLC. Nevertheless, to clarify this point, we have added the following:

- Page 12. “downregulation, consistent with the modelling of PD-1/LAG-3 co-expression signature. We could not evaluate Tregs in our study, as these were below detection in peripheral blood from our NSCLC patient cohort.”

5. The authors focus on YY1 in Figure 3, but DDX21 is also highlighted. Why was YY1 selected? Is DDX21 or any of the other proteins indicated by analysis in Figure 5g associated with response to immunotherapy as in Figure 5j? Also, why did the author look at aggregated T cells rather than at least CD4+ and CD8+ T cells?

We thank the Reviewer for the comment. As mentioned in the manuscript, we obtained a list of 35 targets that were commonly regulated by all immune checkpoints. Some of the up and some of them down-modulated. Therefore, we chose two of them, one up and one down. Several of the others we evaluated through transactivation of their corresponding promoters shown also in figure 3, as a kind of high-throughput assay. YY1 was further selected because there were some publications linking this transcription factor with significant regulatory T cell activities. During the time of the experiments, we studied the potential use of YY1 as a marker of response in immunotherapy, and thus, obtaining valuable data on response. As we had this data from a decent number of patients, we thought that it was highly relevant to publish it in the manuscript. According to presenting the data on the pool of T cells rather than in CD4 and CD8 T cells separately, we do have the data. But YY1 expression between CD4 and CD8 T cells is so similar that we decided to show the whole data to save space. But we believe that this has to be made clearer in the text. The data on DDX21 is limited to just to prove its regulation by flow cytometry, together with the data with the other targets as shown by transcriptional transactivation. Nevertheless, it is important to explain the choice of YY1 and its relationship with response in the text. Therefore, we have added the following:

- Page 10: “From the 35 targets, YY1 has been shown to be key in NF- κ b, ATF, AP-2 α and Myc transcription regulation. Therefore, YY1 was chosen to study its regulation and association with response to immunotherapy.”
- Page 10:” YY1 expression was similar between CD4 and CD8 T cells.”

6. There is a logical jump in Figure 4 that this reviewer finds difficult to follow. The authors identify canonical pathways, upstream regulators and causal networks associated with PD1, LAG3 and PD1+LAG3. Then in Figure 4e the authors present an analysis of E3 ubiquitin ligases with minimal motivation as

to why they are assessing them. They then proceed to investigate CBLs, but CBLs aren't included anywhere in the figure. The authors should better explain the logic of this figure that results in a desire to analyze CBL-B.

We agree with the Reviewer that it is difficult to follow without the proper context, which is not evident in the manuscript text. The data in this manuscript is part of a larger story on LAG-3 signaling. **Again, the reason for this is that to fully explain the choice, we would have to release confidential information in the form of a second paper that we are preparing (with intention to submit to EMBO Mol Med, as a follow-up paper).** And EMBO Mol Med does not allow to cite “data not shown”. More specifically, we have extensive omics data, similar to this study but including about 30 different LAG-3 mutants on the signaling domains. **In our additional data, we directly detect CBL-b and its different phosphorylated isoforms, linking phosphorylation of CBL-b and its regulators with distinct LAG-3 signaling domains.** Most of the data data from this study made us chose CBLB as our target. We would still like to maintain this data confidential for the time being. Again, we offer the possibility of releasing this confidential information to Prof Roth so that she can certify that indeed, we possess these data.

Nevertheless, we can provide an explanation more suited for the purposes of this paper. There are several reasons that we focused our attention on CBLB:

- According to abundant published data including our own (Karwacz et al. 2011. EMBO Mol Med), E3 ubiquitin regulators are key controllers of TCR signal transduction through several means (not only degradation). We therefore evaluated these pathways in our dataset, obtaining results consistent with the pathways of E3 ubiquitin ligases of which CBL-B is a master regulator. But as the referee indicated, although the signature was there, we did not detect CBL-B in the proteomic set belonging to the current manuscript. This is due to a limitation of differential quantitative proteomics: the overall protein coverage, and many proteins are not detected for technical reasons. We knew it was there because CBLB is expressed in Jurkat T cell lines and in primary T cells when appropriately stimulated, as many and others have shown in other papers and in this paper. Again, we have directly detected CBLB and its phosphorylation forms in a larger, follow-up study, by proteomics combined with phosphoproteomics.
- There are highly specific inhibitors available that are currently under clinical evaluation in clinical trials together with PD-1 blockade. That was our decisive factor to target CBLB.

We agree with the Reviewer that the rational should be explained more logically. Therefore, we added the following in our manuscript:

- Page 15: “Our results highlighted a signature associated with E3 ubiquitin ligase pathways in our proteome. However, the coverage of the proteomic database in this study did not include its master regulator CBL-B, although we could detect its expression in both our Jurkat T cell lines and in primary T cells stimulated with cancer cells. As a CBL-B inhibitor is currently under clinical evaluation in combination with anti-PD-1 antibodies (clinical trials.gov ID: NCT05662397), we chose CBL-B as a therapeutic target in combination with PD-1/LAG-3 co-blockade.”.

7. The authors are to be commended for performing analyses in a murine model to evaluate the hypothesis that CBL-B inhibition would improve antitumor immunity. Despite a significant improvement in overall survival with the addition of a CBL-B inhibitor to anti-PD1+anti-LAG3, it would be ideal to understand the cell type(s) responsible for this efficacy. It could potentially be improvement in CD8+ T cell or conventional CD4+ T cell effector function or conversely a reduction in regulatory CD4+ T cell function or both. The authors should address this important mechanistic point.

We agree with the Reviewer, and to address this issue we performed a therapeutic experiment in mice depleted of CD8, CD4 or NK cells (Figure 7). The data clearly showed that therapeutic activity was dependent on CD8 T cells. The data corresponding to the in vivo experiment has been added in the manuscript:

- Page 13: “To identify the main effector cell type for antitumor efficacy in our model, PD-1/LAG-3/CBL-B were blocked in mice depleted of CD8, CD4 or NK cells (Fig 7a). Again, a very highly significant increase in survival was achieved for the triple combination, which more than tripled survival of PD-1/LAG-3 co-blockade alone (Fig 7b). Importantly, only CD8 T cell depletion completely abrogated the anti-tumor efficacy of the triple PD-1/LAG-3/CBL-B blockade combination both in terms of survival (Fig 7b) and tumor growth (Fig 7c-d).”
- Page 20:” For in vivo depletion experiments, CD4, CD8 or NK depletions were carried out by intraperitoneal administration of 100 μ g of anti-mouse CD8a (clone 2.43; BioXCell), CD4 (clone GK1.5; BioXCell) or NK1.1 (clone PK136; BioXCell) anti-bodies following routine techniques in our group (Blanco et al, 2024; Bocanegra et al, 2023). Mice were humanely sacrificed when tumor size reached ~150-200 mm², or when tumor ulceration or discomfort were observed”.
- Figure 7 legend: “Figure 7. CD8 T cells are responsible for the PD-1, LAG-3 and CBL-B triple blockade significant therapeutic antitumour effect. (a) Schematic design of the experiment. BALB/c female mice were randomly

allocated and subcutaneously injected with 2×10^6 Lung adenocarcinoma (Lacun3) cells per animal. 100 μg of anti-PD-1, 100 μg of anti-LAG3, 30mg/kg of CBL-Bi and the corresponding depletion antibodies were administered intraperitoneally at days 0, 2, 6, 9, 13 and 15 as indicated in the figure. NK, CD4, and CD8 T-cell depletions were carried out by intraperitoneal administration of 100 μg of anti-mouse CD8a, CD4 or NK1.1 antibody. Mice were humanely sacrificed when tumour size reached $\sim 150\text{-}200 \text{ mm}^2$, or when tumour ulceration or discomfort were observed. (b) Kaplan-Meier survival plot of mice under the indicated treatments or depletion (percent). Statistical significance was tested with the Log-rank test. (c) Evolution of mean tumour size following the indicated treatments (left). Tumour volumes 9 days after treatment initiation (right). Error bars correspond to $\pm\text{SEM}$ (left) and box and whiskers with min to max values (right) ($n=6$ mice per group). Data information: Statistical comparisons were carried out by a two-way ANOVA followed by pair-wise Tukey tests. (d) Tumour growth of individual mice in the indicated treatment groups ($n=6$ mice per group). Statistical comparisons are shown in the graph as indicated in Materials and Methods. Data information: Briefly, for (b), survival was represented by Kaplan–Meier plots and analyzed by log-rank test. For (c), statistical comparisons were carried out by a two-way ANOVA followed by pair-wise Tukey tests. *, **, ****, indicate significant ($P<0.05$), very significant ($P<0.01$) and highly significant ($P<0.0001$) differences.”

8.Minor Comments

1. Line numbers would be helpful for calling out specific corrections.

We agree with the Reviewer and have included line numbers in the reviewed version of the manuscript.

2. Page 4, first paragraph: "Hence, PD-1 and LAG-3 cooperatively establish a strong dysfunctional estate", estate should be "state".

We have corrected it as indicated.

3. Page 4, first paragraph: "PD-1 and LAG-3 stablish a complex together", stablish should be "establish".

We have corrected it as indicated.

4. Page 14, first paragraph: "This evidenced the association of the PDCD1/LAG3signature with potentially", evidenced should be "demonstrated" or something similar.

We have substituted evidenced with demonstrated.

5. Page 14, second paragraph: "The engineering of PD-1+LAG-3 T-cell lines uncovered several T-cell dysfunctions associated to this signature." "To" this signature should be "with" this signature.

We have corrected it as indicated.

6. The last paragraph of the discussion is unnecessary and could be mentioned elsewhere in the discussion. It would be stronger to end on the preceding paragraph.

We agree with the Reviewer and we have removed the last paragraph as it does not provide critical or novel information.

7. Methods, under flow cytometry: several of the Greek letters do not appear and are instead replaced by empty boxes.

Corrected as indicated.

8. Mention of "tox" lists is confusing, given the role of the gene TOX in CD8+ T cell exhaustion.

We agree with reviewer that the mention of tox lists is confusing, so the diseases and bio functions / and tox lists data are now plotted as "Molecular and Cellular functions" and "Diseases and disorders" figure.

9. The authors erroneously refer to Figure 5g as Figure 1g in the results.

Corrected as indicated. Fig 1g reference refers to the modelling of PD-1/LAG-3 co-expression signature that was previously presented on the previous results sections. We apologize to the reviewer for the redundancy, so reference to the figure in this later section has been removed.

4th Jun 2024

Dear Dr. Escors,

Thank you for submitting your revised study, which was sent back to referees #1 and #3. As you will see below, the referees are overall satisfied with the revisions, and I will therefore be able to accept your manuscript once the following points will be addressed:

1/ Referees' comments: please address the remaining minor concerns from the referees (no further reaching experiments requested).

2/ Manuscript text:

- Please remove the highlighted font, and only keep in track changes mode any new modification.
- Authors: discrepancies between the submission system and the manuscript were found in the following authors names: Maria Jesus Garcia / Maria Jesus Garcia-Granda and Gonzalo Fernandez / Gonzalo Fernandez-Hinojal; the following author emails bounced: miren.zuazo.lbarra@navarra.es, gonzalo.fernandez.hinojal@navarra.es, ai.bocanegra.gondan@navarra.es, jlegg@crescendobiologics.com.
- Methods:
 - o Cells: please indicate whether the cells were authenticated and tested for mycoplasma contamination.
 - o Antibodies: please provide concentration/dilutions
 - o Mice: please indicate the age of the animals at the time of experiment, as well as the housing and husbandry conditions.
- Data availability: please move this section to the end of the Methods section:
 - o Please note that the specific URLs for PXD040408, JPST002062 datasets are not provided in the data availability statement.
 - o Please note that reviewer access code for JPST002062 dataset provided in the manuscript is incorrect.
 - o Please note that the datasets must be made public before acceptance of the manuscript
- Acknowledgements: please make sure that the information provided matches the information entered in the submission system (currently, Spanish Association against Cancer (AECC), PROYE16001ESC; TRANSPOCART ICI19/00069; Strategic projects from the Department of Industry, Government of Navarre (AGATA, Ref. 0011-1411-2020-000013; LINTERNA, Ref. 0011-1411-2020-000033; DESCARTHES, 0011-1411-2019-000058); European Union Horizon 2020 ISOLDA project, under grant agreement ID: 848166 and Navarrabiomed-Fundación Miguel Servet predoctoral contract are currently missing in the submission system).
- Author contributions: CRediT has replaced the traditional author contributions section because it offers a systematic machine-readable author contributions format that allows for more effective research assessment. Please remove the Authors Contributions from the manuscript and use the free text boxes beneath each contributing author's name in our system to add specific details on the author's contribution. More information is available in our guide to authors.
- Please rename "Conflicts of interests" to "Disclosure statement and competing interests" section should be included after the acknowledgements (<https://www.embopress.org/competing-interests>).

3/ Figures:

- Please make sure that all figures and figure panels are referenced in the manuscript text (a callout is missing for Fig 5F).
- Please address the queries from our data editors in the figure legends:
 1. Please note that the exact p values are not provided in the legends of figures 2d-f; 3h; 5b-e; 6b, e-f; 7b-c.
 2. Please indicate the statistical test used for data analysis in the legends of figures EV 3b-c.
 3. Please note that in figures 2e; 5c; 7b-c; there is a mismatch between the annotated p values in the figure legend and the annotated p values in the figure file that should be corrected.
 4. Please note that the box plot needs to be defined in terms of minima, maxima, centre, bounds of box and whiskers, and percentile in the legend of figure 3h.
 5. Please note that the box plots need to be defined in terms of centre, bounds of box and percentile in the legends of figures 5c-e; 6b, e; 7c.
 6. Please note that information related to n is missing in the legend of figure 6a.
 7. Although 'n' is provided, please describe the nature of entity for 'n' in the legends of figures 2d-f.
 8. Please note that the measure of center for the error bars needs to be defined in the legends of figures 2d-f; 5b; 6a.
 9. Please note that scale bar and its definition are missing for figure 2c.
- Appendix Tables S1 and S2 should be renamed Dataset EV1 and EV2. Both files need a legend added in a separate worksheet.

4/ Thank you for providing Source Data. Please upload them as one file per figure, and upload also the completed source data checklist. Please check the values provided in the following files: Figure 3j (see attached file).

5/ Checklist:

Please fill in the "Experimental study design and statistics"/"inclusion/exclusion criteria" section.

6/ Thank you for providing a nice synopsis picture. Please resize it to a 550 px wide x 300-600 px high file, and make sure that the text remains legible.

Please also provide a synopsis text, which should include a short stand first (maximum of 300 characters, including space) as well as 2-5 one-sentences bullet points that summarizes the paper (maximum of 30 words / bullet point).

7/ As part of the EMBO Publications transparent editorial process initiative (see our Editorial at <http://embomolmed.embopress.org/content/2/9/329>), EMBO Molecular Medicine will publish online a Review Process File (RPF) to accompany accepted manuscripts.

This file will be published in conjunction with your paper and will include the anonymous referee reports, your point-by-point response and all pertinent correspondence relating to the manuscript. Let us know whether you agree with the publication of the RPF.

I look forward to receiving your revised manuscript.

Yours sincerely,

Lise Roth

***** Reviewer's comments *****

Referee #1 (Remarks for Author):

Although for non specialists, the work remains somewhat difficult to read due to its complexity, the authors have made an effort to simplify the figures (even if most of them remain very dense), answer questions raised by the reviewers and clarify some points.

A few comments:

- L56: established
- Why are the Fig3b canonical pathways different from the canonical pathways of Fig3C in the 1st version?
- In the legend, the gray round dots correspond to "No activity pattern". Unless I've misunderstood something, is it normal to have for Top 10 enriched
- Fig. 3d : causal networks are missing
- Why are the Fig. 4a version 2 canonical pathways and upstream regulators different from the canonical pathways and upstream regulators of Fig4a in the 1st ?
- Lane 320 : PD1+/LAG3+ infiltrating cells are predominantly CBL Bneg since CLB B+ represent only 22% of PD1+/LAG3+.
What is the impact of CBL Bneg in ICI resistance?
- Fig. 6: black band at the bottom of the figure (unless the problem is mine).
- L376 : diameter =3.5 mm not mm2
- L414 : remains controversial

Referee #3 (Comments on Novelty/Model System for Author):

Model system is acceptable.

Referee #3 (Remarks for Author):

The authors have undertaken substantial efforts to revise this manuscript in a relatively short period of time. Especially nice was the mechanistic demonstration that CD8+ T cells are required for the efficacy of anti-PD1 + anti-LAG3 + CBL-B inhibition. This reviewer appreciates that the authors have additional data that they would like to use for another manuscript, but the phosphoproteomic data the authors reference with regards to CBL-B are important to the overall message of this manuscript. If

the authors wish to not include this data and make the logical leap to CBL-B from their bioinformatic analyses, they should i) include some type of enrichment score and statistical significant for E3 ligase family enrichment associated with Figure 4c and ii) look at expression level of the all E3 ubiquitin ligase genes in PDCD1+LAG3+ T cells from the scRNAseq data in Figure 5. This would bring up CBL-B amongst other targets, and the authors can then mention that CBL-B inhibitors are under clinical development. Finally, the authors could have readily performed targeted assays such as flow cytometry for phosphorylated proteins to assess differences in phosphorylation of proteins involved in the TCR signaling cascade without compromising other data for a subsequent manuscript. Overall, the authors show a strong phenotype with CBL-B inhibition in conjunction with anti-PD1 and anti-LAG3 in mice, but the logic of the bioinformatics to arrive at CBL-B could still be further improved.

Recommendation: Minor revisions

REFEREE 1

Although for non specialists, the work remains somewhat difficult to read due to its complexity, the authors have made an effort to simplify the figures (even if most of them remain very dense), answer questions raised by the reviewers and clarify some points.

We sincerely appreciate the comments from Referee 1. Indeed, the paper is complex, but the information is proving to be invaluable for us. We are convinced that it will also be of high importance to researchers in both medical oncology, pharmacology and basic immunooncology.

A few comments:**1. L56: established**

Thank you. “stablished” has been changed by “established” as indicated.

2. Why are the Fig3b canonical pathways different from the canonical pathways of Fig3C in the 1st version?

In the process of revision, the IPA software version was updated and data was rerun with this later version to provide the most updated and accurate results at the final version of the manuscript. However, this does not impact nor change the main message that can be extracted from the results, as the altered molecular processes

Navarrabiomed-FMS
C/Irunlarrea 3, 31008 Pamplona, Navarra, Spain
descorsm@navarra.es

obtained from the analyses remain to be the same. To clarify this to the reader, the following has been added to the figure legends, materials and methods and manuscript text: “(accessed on 2024)”

3. In the legend, the gray round dots correspond to "No activity pattern". Unless I've misunderstood something, is it normal to have for Top 10 enriched

Thank you very much for raising this question. IPA software was used for these analyses, that utilizes z-score to assess the match of observed and predicted regulation patterns. Given the observed differential regulation of a gene (“up” or “down”, based on Log_2 FoldChange data) in the input dataset, IPA calculates directionality (z-scores). For figures 3b and 3d that present grey dots (“no activity pattern”), Perseus ANOVA multiple sample test was used to obtain the significant input data that was used for the IPA analysis. As this type of analysis does not include direct comparisons between two groups, Perseus does not calculate Log_2 FoldChange, and thus IPA cannot calculate z-scores.

4. Fig. 3d: causal networks are missing

We appreciate the comments from the Reviewer, and we acknowledge that the final version could have not been possible without the Reviewers assistance. Yes indeed, the information is complex and large, but rather of major importance to the field. It was truly an effort to reduce all the information to the figures shown in the paper, as Reviewer 1 rightly said in the first round of revision, figures could be found difficult to read and with too many information, sometimes redundant. Hence, we took on board the Reviewer’s suggestion of reconsidering whether each panel of each figure is essential to understanding and how the figure can best convey the desired information, and we went through every figure one by one, and undertook steps to simplify them. This included removing from each figure the panels less important or redundant panels as suggested. Thus, the causal networks were found to be redundant with upstream regulators, and hence, only upstream regulators remain in the final figure. We believe that the simplified version of the figures will greatly help to get the important information at first glance.

5. Why are the Fig. 4a version 2 canonical pathways and upstream regulators different from the canonical pathways and upstream regulators of Fig4a in the 1st?

In the process of revision, the IPA software version was updated and data was rerun with this later version to provide the most updated and accurate results at the final version of the manuscript. However, this does not impact nor change the main message that can be extracted from the results, as the altered molecular processes obtained from the analyses remain to be the same. To clarify this to the reader, the following has been added to the figure legends, materials and methods and manuscript text: “(accessed on 2024)”.

6. Line 320 : PD1+/LAG3+ infiltrating cells are predominantly CBL Bneg since CLB B+ represent only 22% of PD1+/LAG3+. What is the impact of CBL Bneg in ICI resistance?

Thank you very much for raising this question. According to our experience, 22% of total infiltrating T cells is by no means a small value, but rather large. We can speculate on the role of CBLB neg PD1+/LAG3+, which we have added in the discussion section. Most likely, these T cells are activated T cells that infiltrate into the tumor, as T cell activation leads to upregulation of PD1 and LAG3. Most likely these T cells have not been inactivated yet by the tumor, and may be exerting anti-tumor effects. We have to remember that the relationship between the tumor and immune cells is dynamic, and phenotypes will likely change with time, and also by immune editing.

7. Fig. 6: black band at the bottom of the figure (unless the problem is mine).

It is an unrelated problem from the submission system, as the figure itself is correct. If the figure is opened directly on the computer, that black band disappears.

8. L376 : diameter =3.5 mm not mm2

Thank you. “mm2” has been changed by “mm” as indicated.

9. L414 : remains controversial

Thank you. “remains to be controversial” has been changed by “remains controversial” as indicated.

REFEREE 3

The authors have undertaken substantial efforts to revise this manuscript in a relatively short period of time. Especially nice was the mechanistic demonstration that CD8+ T cells are required for the efficacy of anti-PD1 + anti-LAG3 + CBL-B inhibition.

We appreciate the comments from the Reviewer, and we acknowledge that the final version could have not been possible without the Reviewers assistance.

1. This reviewer appreciates that the authors have additional data that they would like to use for another manuscript, but the phosphoproteomic data the authors reference with regards to CBL-B are important to the overall message of this manuscript. If the authors wish to not include this data and make the logical leap to CBL-B from their bioinformatic analyses, they should i) include some type of enrichment score and statistical significant for E3 ligase family enrichment associated with Figure 4c and ii) look at expression level of the all E3 ubiquitin ligase genes in PDCD1+LAG3+ T cells from the scRNAseq data in Figure 5. This would bring up CBL-B amongst other targets, and the authors can then mention that CBL-B inhibitors are under clinical development.

We appreciate the Reviewers comments. We faced the problem that if we did include phosphoproteomic data of wild-type proteins and mutants, the proteomic dataset would have to be publically released at this stage. This will seriously compromise the second paper, because we are in the process of finishing experiments, which include also *in vivo* studies. We could have also solved this issue by stating the data that we have with “Results not shown” or “in preparation”, but I am afraid that EMBO policy does not allow to do it. Hence, we have followed the Reviewers’ suggestion and extracted an E3 ubiquitin ligase related gene expression profile of more than 100 genes on the *PDCD1+* and *LAG3+* CD4, CD8, Treg and NK infiltrating population on the NSCLC tumor microenvironment, including *ANAPC11*, *APC*, *BARD1*, *BIRC7*, *BRCA1*, *BRCA2*, *CBL*, *CBLB*, *CBLC*, *CDC34*, *COP1*, *CRBN*, *CRY2*, *CUL1*, *CUL2*, *CUL3*, *CUL4A*, *CUL4B*, *CUL5*, *CUL7*, *CUL9*, *DDB1*, *DDB2*, *FBXL3*, *FBXL4*, *FBXL5*, *FBXL6*, *FBXL7*, *FBXL8*, *FBXO2*, *FBXO38*, *FBXO4*, *FBXW7*,

HECTD1, HECTD2, HECTD3, HECTD4, HUWE1, ITCH, KCMF1, KEAP1, KLHL22, MDM2, MIB1, NEDD4, PPARG, RBX1, RING1, RNF10, RNF11, RNF114, RNF126, RNF13, RNF138, RNF14, RNF148, RNF168, RNF17, RNF2, RNF20, RNF38, RNF4, RNF40, RNF5, RNF6, RNF7, RNF8, SKP1, SKP2, SPOP, TRAF2, TRAF6, TRIM25, TRIM63, UBA1, UBA2, UBA3, UBA5, UBA6, UBA7, UBAP1, UBAP1L, UBAP2, UBAP2L, UBE2C, UBE2D1, UBE2D2, UBE2D3, UBE2D4, UBE2K, UBE2N, UBE2S, UBE3C, UBE4A, UBE4B, UBF1, UCHL1, UCHL3, UCHL5, UFM1, USP14, USP48, VHL, WWP1 and XIAP.

We have therefore added this database and the following sentence in page 12.

CBLB was brought up among more than 100 ubiquitin-related target genes in lymphoid infiltration, and it was found to be among the top E3 ubiquitin ligases co-expressed with *PDCD1* and *LAG3* in the tumor microenvironment (**Dataset EV3**).

2. Finally, the authors could have readily performed targeted assays such as flow cytometry for phosphorylated proteins to assess differences in phosphorylation of proteins involved in the TCR signaling cascade without compromising other data for a subsequent manuscript.

Thank you very much for the suggestion, but following the same reasoning from the previous point, we rather publish that data in the context of our next paper. Nevertheless, at this stage it would not have made a significant impact in the message from the current manuscript.

3. Overall, the authors show a strong phenotype with CBL-B inhibition in conjunction with anti-PD1 and anti-LAG3 in mice, but the logic of the bioinformatics to arrive at CBL-B could still be further improved.

We appreciate the comment from the Reviewer, and we are aware of this. Unfortunately, disclosing the full experimental data and information leading to CBLB would imply releasing of the dataset from a second following paper. We have therefore, made an effort to clarify this issue following the guidelines suggested by Reviewer 3.

Recommendation: Minor revisions

21st Jun 2024

Dear Dr. Escors,

Thank you very much for your prompt reply in submitting the revised files. I am pleased to inform you that your manuscript is now accepted for publication!

Please note that I have added the following sentence in the legend of Fig 2c: "Scale bars and inserts are indicated in the original pictures in the Source Data file for this figure panel." Let us know immediately if you do not agree.

Please also provide the Appendix as a PDF file, with title, table of content and page numbers. You may send this file via email, we'll upload it in the system.

Your manuscript will then be sent to our publisher to be included in the next available issue of EMBO Molecular Medicine.

If you have any questions, please do not hesitate to contact the Editorial Office.

Congatulation on your work!

With kind regards,

Lise Roth
